# Bacterial nanotubes as a manifestation of cell death

Jiří Pospíšil [1], Dragana Vítovská [1], Olga Kofroňová [2], Katarína Muchová [3], Hana Šanderová [1],
Martin Hubálek[4], Michaela Šiková [1], Martin Modrák [5], Oldřich Benada [2✉], Imrich Barák [3✉] &
Libor Krásný [1✉]

Bacterial nanotubes are membranous structures that have been reported to function as
conduits between cells to exchange DNA, proteins, and nutrients. Here, we investigate the
morphology and formation of bacterial nanotubes using *Bacillus subtilis*. We show that
nanotube formation is associated with stress conditions, and is highly sensitive to the cells'
genetic background, growth phase, and sample preparation methods. Remarkably, nanotubes
appear to be extruded exclusively from dying cells, likely as a result of biophysical forces.
Their emergence is extremely fast, occurring within seconds by cannibalizing the cell
membrane. Subsequent experiments reveal that cell-to-cell transfer of non-conjugative
plasmids depends strictly on the competence system of the cell, and not on nanotube for-
mation. Our study thus supports the notion that bacterial nanotubes are a post mortem
phenomenon involved in cell disintegration, and are unlikely to be involved in cytoplasmic
content exchange between live cells.

[1] Laboratory of Microbial Genetics and Gene Expression, Institute of Microbiology of the Czech Academy of Sciences, 142 20 Prague 4, Czech Republic.
[2] Laboratory of Molecular Structure Characterization, Institute of Microbiology of the Czech Academy of Sciences, 142 20 Prague 4, Czech Republic.
[3] Department of Microbial Genetics, Institute of Molecular Biology, Slovak Academy of Sciences, 845 51 Bratislava, Slovakia. [4] Institute of Organic Chemistry
and Biochemistry of the Czech Academy of Sciences, 160 00 Prague 6, Czech Republic. [5] Laboratory of Bioinformatics/Core Facility, Institute of Microbiology
of the Czech Academy of Sciences, 142 20 Prague 4, Czech Republic. ✉email: benada@biomed.cas.cz; imrich.barak@savba.sk; krasny@biomed.cas.cz

Bacteria are one of the most dominant forms of life on Earth, employing a vast range of strategies to exploit their environments. One of these strategies may involve formation of tubular membranous structures. These structures are termed nanotubes (NTs) in *Bacillus subtilis* and *Escherichia coli*[1,2], nanowires (NWs) in *Shewanella oneidensis*[2–4], nanopods (NPs) in *Delftia* sp. Cs1-4 and in hyperthermophilic archaea of the genus *Thermococcus*[5–7], or outer membrane tubes (OMTs) in *Myxococcus xanthus*[8]. These structures are composed of cytoplasmic or outer membranes, depending on the species of origin, and may serve different functions in these organisms. The NTs of *B. subtilis* are perhaps the best characterized example. They were reported to frequently occur in exponentially growing cells: ~70% of cells contained NTs and a single cell contained several of them[9].

YmdB, a phosphodiesterase that hydrolyzes cyclic nucleotides such as cAMP[10], and flagellar body proteins[9,11] have been reported to be necessary for NT formation in *B. subtilis*. YmdB was found to localize to NTs[11], though the exact mechanism by which it contributes to NT formation is unknown. The flagellar body proteins required for the flagellar export apparatus, called CORE, function both in flagella and in NT assembly[9,12].

Two classes of *B. subtilis* NTs have been recognized: (i) extending nanotubes (attached to a single cell) and (ii) intercellular nanotubes (connecting two cells)[1,13]. Extending NTs are thought to increase the surface area of the cell and contribute to nutrient uptake. Intercellular NTs can function as conduits for transport of molecules such as metabolites (e.g., amino acids), proteins (including toxins), and even non-conjugative plasmids[1,2,14]. These intercellular tubes can be formed between two cells of a single bacterial species, between cells of two different bacterial species, and even between a bacterium and a eukaryotic host, where the bacterium uses NTs to extract nutrients from its host, as reported for enteropathogenic *E. coli*[15].

NTs appear to be a common phenomenon shaping intra-, and inter-species interactions, with far reaching consequences for our understanding of, and fight against bacterial pathogens[16]. Yet, the number of reports on these structures is relatively small, and models of their functioning are still poorly defined. Therefore, detailed insights into this cellular feature are needed.

Here, we focus on *B. subtilis* NTs and identify genes and conditions required for NT formation. We show that under non-stress conditions, NTs are rare; under stress, the number of NTs increases. Most importantly and surprisingly, these structures are formed when cells are dying or even after cell death and, therefore, they are unlikely to be involved in nutrient uptake or cytoplasmic content exchange as proposed by previous studies. This is demonstrated by the complete absence of non-conjugative plasmid transfer in a Δ*comK* strain, which is still able to form NTs [ComK is essential for bacterial competence and DNA uptake[17]]. The results of this study, therefore, indicate that NTs are an attribute of *dying* cells and are not involved in the exploitation of the environment by *live* cells.

## Results

**Identification of NTs**. Initially, we wished to detect NTs in *B. subtilis*. Using a combination of scanning electron microscopy (SEM) and structured illumination microscopy (SIM), we analyzed wt *B. subtilis* cells (BSB1) grown to exponential phase in liquid LB. The electron micrographs revealed that at least two types of filamentous structures were present: (i) numerous thinner filaments (diameter < 30 nm) and (ii) rare thicker filaments (diameter ~70 nm). Since *B. subtilis* can form flagella, we also examined a Δ*hag* strain lacking the gene encoding flagellin, the principal flagellar component[18]. Supplementary Fig. 1a–c clearly

shows the disappearance of the numerous thinner filaments in the Δ*hag* strain; we therefore surmised that the remaining thicker filaments were NTs. In a subset of these filaments we observed elongated, flattish terminal structures (Supplementary Fig. 1d, e). The frequency of NTs was rather low: only one NT approximately per 500 cells.

To verify the NT identification, we used SIM to examine a strain bearing a single amino acid substitution in flagellin [hag^T209C][19], which allows this protein to be stained with a maleimide derivative of Alexa Fluor 488, thereby distinguishing it from membranous structures stained with Nile Red. Supplementary Fig. 1g shows one cell with a large number of flagella and a single membranous structure, a nanotube, which was the typical number of NTs per cell (when present). The rare terminal structures were also occasionally detected by this method. It is important to note that the SIM approach required no cell fixation (unlike SEM where it is an integral part of the protocol), thereby excluding the possibility that these structures were byproducts of the cell fixation procedure. Nevertheless, the NT-bearing cells displayed patchy staining with Nile Red, which may indicate non-optimal cell conditions. Taken together, we had thus succeeded in identifying NTs, although the frequency of their occurrence was significantly lower than that claimed previously[1,9].

**Genetic requirements for NT formation**. We next determined which genes were required for NT formation. Although several genes had already been reported, we took a systematic, unbiased approach, and used a number of strains with deletions in one or more sigma factors. These factors associate with RNA polymerase (RNAP), which is responsible for the transcription of DNA into RNA, and provide the holoenzyme with an affinity for specific DNA promoter sequences[20]. Using this approach, we wanted to identify the regulon that contains genes necessary for NT formation. Altogether, we tested deletions of 10 alternative sigma factors [out of the 19 present in *B. subtilis*[21,22]] and monitored NT formation by SEM. Of the 10 tested sigma factors, only SigD[23,24] was required for NT formation (Supplementary Fig. 1h–l). However, due to the extremely low numbers of NTs that can be found by SEM under these conditions, the results were not statistically significant. Nevertheless, the absence of NTs in the Δ*sigD* strain was consistent with the known NT requirement for the CORE proteins, whose genes are SigD-dependent[25].

**Conditions under which the majority of cells form NTs**. Both SEM and SIM are high-resolution imaging techniques that are best suited for capturing static structures. To gain more detailed insights into the dynamics of NT formation and their potential movements, we therefore used time-lapse imaging with a fluorescence microscope. In Fig. 1a we used 1× phosphate-buffered saline (PBS) agar pads covered with coverslips ("Glass-Agar-Glass"—the GAG method). At times $t = 0$ min and $t = 15$ min, no NTs were detectable in wt *B. subtilis* cells from exponential phase, consistent with the low frequency of NT formation we had observed by SEM and SIM. However, when we used glass slides and coverslips coated with poly-L-lysine and the bacteria in liquid 1 × PBS ("Glass-Liquid-Glass"— the GLG method), a few NTs were detected. We noticed, though, that when we firmly pressed down the coverslip to obtain a monolayer of bacteria (a common technique to prepare samples), the number of NTs increased as opposed to when the coverslip was gently positioned over the sample. Therefore, we applied defined pressure (P; ~80 kPa) on the coverslip (P-GLG), and this resulted in a large number of NTs originating from many cells within 15 min; the number of NTs approached those previously reported (Fig. 1b and Supplementary Movie 1)[9]. The same result was obtained regardless of the wt

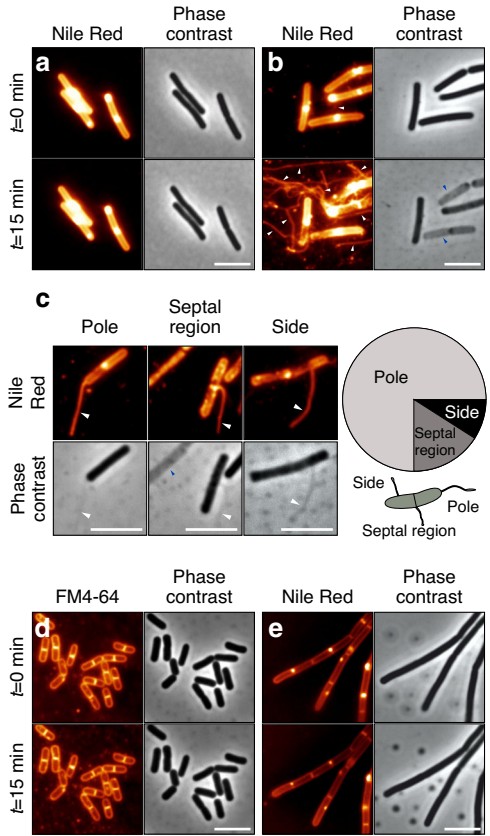

**Fig. 1 NT formation in exponential and stationary phase.** These images were acquired by fluorescence microscopy; the corresponding phase-contrast images are also shown. For definition of $t = 0$ refer to Mat & Met. **a** Exponentially grown wt *B. subtilis* (LK1432) cells prepared by the GAG method. **b** Exponentially grown wt *B. subtilis* (LK1432) cells prepared by the P-GLG method. **c** One-hundred exponentially growing wt *B. subtilis* (LK1432) cells bearing NTs were analyzed. Most NTs emanated from the cell poles (75 NTs), followed by NTs from the septal region (16 NTs) and the sides (9 NTs). The positions of NT attachment to the rod-shaped cells were not uniformly distributed over the cell surface ($p < 0.001$, custom test —see Quantification and statistical analysis). **d** Stationary phase wt *B. subtilis* (LK1432) cells prepared by the P-GLG method but to stain the membranes, FM4-64 was used instead of Nile Red, as Nile Red poorly stains membranes of cells from this phase (Supplementary Fig. 2d). **e** Exponentially grown Δ*sigD* (LK1873) cells prepared by the P-GLG method. The images were taken at 0 and 15 min time points. White arrows indicate NTs, blue arrows indicate ghost cells in the phase-contrast images. The membrane stain (false colored glow, FM4-64 or Nile Red) is indicated above the images. Scale bar = 5 μm. All experiments (**a**–**e**) were conducted in at least three biological replicates with similar results.

strain used: PY79 behaved like BSB1 (Supplementary Fig. 2a–c). The positions of NT attachment to the rod-shaped cells were not random: the majority of NTs (75%) originated from the cell poles, 16% from septal regions and 9% from other regions (Fig. 1c, p-value < 0.001). Moreover, in the phase-contrast images we noticed that the NTs were in some cases attached to gray cells (Fig. 1b, right hand panel, and Supplementary Fig. 2a–c). These cells are characterized by a decrease in phase-contrast due to the efflux of cytoplasmic contents. This is typically associated with cell death[26–28] and we refer to such cells as ghost cells.

In parallel, we performed experiments with wt stationary phase cells using the P-GLG experimental setup. These cells did not produce NTs within the 15 min time course (Fig. 1d). The absence of NTs from wt stationary phase cells might be due to the

thicker and more ordered cell wall of non-dividing cells[29], presenting a stronger barrier to NT extrusion. We also viewed exponential phase Δ*sigD* cells using the same approach, and these also produced no NTs within the 15 min time course (Fig. 1e). We then speculated that other genes from the SigD regulon besides the CORE genes might also be involved in NT formation. Obvious candidates are autolysins - peptidoglycan hydrolases that open meshes in the peptidoglycan net allowing the insertion of newly synthesized material for surface expansion and cell separation[30]. LytE and LytF are major *B. subtilis* cell wall hydrolases that localize to the septal regions and the cell poles[31]. Consistent with the hypothesis, Δ*lytEF* deletion strain arranged cells into chains, but only during exponential phase (Supplementary Fig. 3a, b). In stationary phase, other autolysins must be responsible for cell separation. Supplementary Fig. 3c shows that exponential phase Δ*lytEF* cells behave like Δ*sigD* cells within the 15 min time-lapse experiment, consistent with the hypothesis that the weakening of peptidoglycan that occurs during its remodeling may be instrumental in NT extrusion. A mass spectrometry comparison of the membrane fractions of wt and Δ*sigD* cells subsequently confirmed that the two strains differed in the presence of autolysins, including LytF (Supplementary Fig. 3d).

Taken together, the experiments performed thus far indicated that the frequency of NT formation depended on (i) the genetic background of the cell, (ii) the method of sample preparation, and (iii) the growth phase of the culture. Intriguingly, phase-contrast images indicated that some of the NT containing cells (ghost cells) might be dying.

Therefore, we added the SYTOX Green stain (SYTOX) to the cells and prepared the samples by the GLG or P-GLG method. SYTOX penetrates into the cells with compromised/permeabilized membranes where it stains nucleic acids and indicates cell death[32]. Figure 2a shows respective SEM and fluorescence microscopy images. The SEM images illustrate the severe damage caused to cells by the P-GLG experimental setup. Subsequent analysis revealed that NTs were associated exclusively with SYTOX-stained cells (Fig. 2b, c). Moreover, the P-GLG method resulted in larger amounts of dying cells and NTs than the GLG method (Fig. 2b, c, $p < 0.001$ [numbers of dying cells between P-GLG and GLG], two-sided, $z = 10$, 95% CI OR = 11–37 and $p = 0.04$, two-sided, $z = 2.05$, 95% CI OR = 1.2–16 [numbers of NTs between P-GLG and GLG]; both GLM [Generalized Linear Model], two-sided; Supplementary Fig. 4a). To further confirm this, we examined those areas where the cells were partially under the coverslip and partially only under the immersion oil. We detected a clear boundary: the cells under the coverslip (the P-GLG method) were frequently SYTOX-stained whereas cells in their immediate vicinity but not under the coverslip were mostly SYTOX-free (Supplementary Fig. 4b). This also shows that poly-L-lysine, which was previously reported to negatively impact bacterial growth[33], is not a major factor causing SYTOX permeability. We also noted that over time, as the cells were progressively losing their contents, including DNA, the SYTOX signal decreased or even disappeared as the SYTOX-stained DNA leaked out of the dead cell.

**NTs are formed from dying cells.** Next we asked whether NTs form before or after cell death. To answer this, we performed a detailed study of the kinetics of NT formation using cells from exponential phase and the P-GLG experimental setup. Figure 3a, c shows that NTs are formed from wt cells within seconds after SYTOX entry. The decrease in signal density in the phase-contrast panel should also be noted. The order of events (SYTOX green penetration, formation of NTs, DNA release, and ghost cell appearance) is shown in a time-lapse microscopy movie

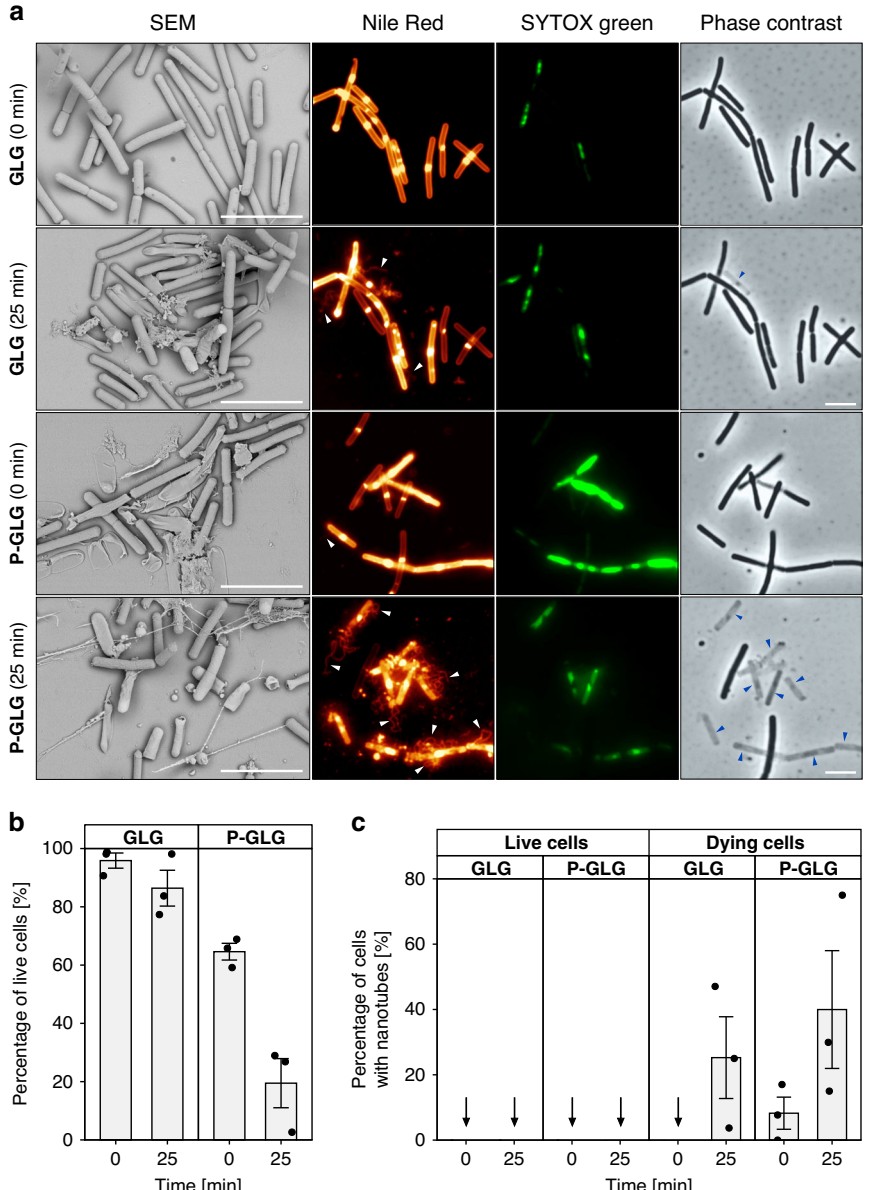

**Fig. 2 Effect of coverslip pressure on NT formation. a** SEM-exponentially growing *B. subtilis* Δ*hag* (LK1966) cells spotted on an uncoated glass slide and covered with a poly-L-lysine coverslip. The coverslips were either gently positioned over the sample (the GLG method) or pressed down (the P-GLG method). $t = 0$, the coverslip, after being positioned over the sample, was immediately removed (for SEM), or the sample, still with the coverslip, was immediately imaged (fluorescence microscopy); $t = 25$, the same with $t = 0$ but the coverslip was left over the sample for 25 min. Before SEM, bacteria were fixed. For fluorescence microscopy, non-fixed living bacteria (LK1432-wt) were observed. White arrows indicate NTs, blue arrows indicate ghost cells. Membranes were stained with Nile Red (false colored glow). Dying cells were monitored with SYTOX green (green). Scale bars in all images represent 5 μm. All experiments were conducted in three biological replicates with similar results. **b** Effect of sample preparation methods on bacterial viability. Low pressure (GLG) and high pressure (P-GLG) methods were compared at two different time points ($t = 0$ and $t = 25$ min). Live and dying cells were quantified according to the presence or absence of the SYTOX signal within the cells. We note that among live cells may be also those that are actually dead and intact, impenetrable to SYTOX. Significantly fewer live cells were observed by P-GLG (at both times) than by GLG ($p < 0.001$, GLM, two-sided, $z = -10$, 95% CI OR = 0.02–0.08) and at the later time points for both methods ($p < 0.001$, GLM, two-sided, $z = -4$, 95% CI OR = 0.12–0.45). For both conditions, a total of 400 cells from three independent experiments were analyzed (the black dots represent individual replicates). The bars are averages and error bars ± SEM. **c** The presence/absence of NTs in samples that were prepared by two different methods (GLG or P-GLG method). Quantification of NTs was based on the Nile Red signal. The percentage of NTs is expressed relative to the number of dying (SYTOX positive) or live (SYTOX negative) cells, which was set as 100% (e.g., out of the 10% of dying cells, 20% formed NTs—that is 2% of the total number of cells). The P-GLG method of sample preparation (compared to GLG) resulted in larger amounts of dying cells with NTs at both time points ($p = 0.04$, GLM, two-sided, $z = 2.05$, 95% CI OR = 1.2–16). No NTs were detected in live cells. For both conditions, 400 cells from three independent experiments were analyzed (the black dots represent individual replicates). The vertical arrows in the chart indicate values "zero". The bars are averages and error bars ± SEM (standard error of the mean).

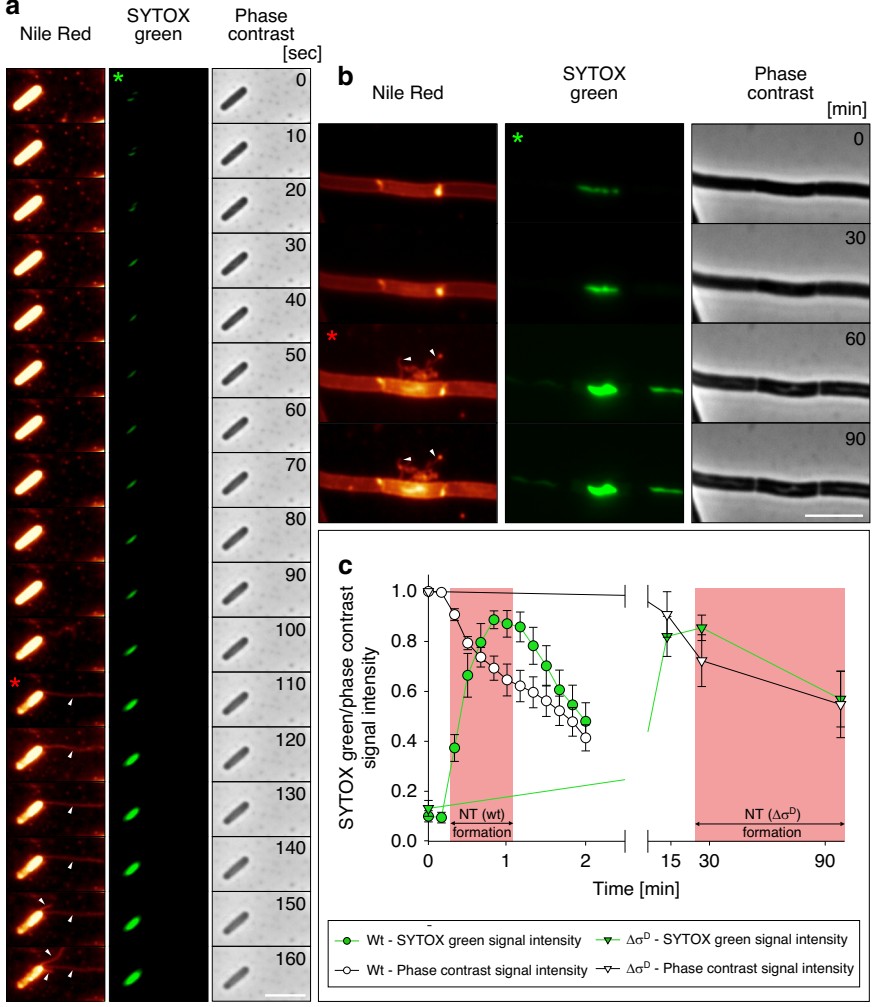

**Fig. 3 Correlations between NT formation and cell death. a** Exponentially growing wt *B. subtilis* (LK1432) cells stained with Nile Red (false colored glow) and SYTOX green (green). Bacteria were prepared by the P-GLG method. The cells were observed every 10 s. Three channels of the same cell are shown: Nile Red, SYTOX green, and phase-contrast. White arrows indicate NTs. The numbers in the phase-contrast panels are seconds. Notice the progressive loss of phase-contrast of the cells as the time advances. In panels **a**, **b** the green asterisk indicates the time when SYTOX green was inside the cell (i.e., the cell was dying). The red asterisk indicates the start of NT formation. **b** Δ*sigD* (LK1873) cells were prepared in the same manner as wt in **a**. White arrows indicate NTs. Numbers in the phase-contrast panels are minutes. Scale bar = 5 μm. **c** Kinetics of NT formation and in wt (LK1432) and Δ*sigD* (LK1873) strains. NT formation occurs in dying cells of both strains, however, the kinetics of SYTOX green penetration and NT formation differs between these strains; it takes ~30x longer in Δ*sigD*. Symbol definitions are shown below the graph. The pink area marks the time limits of NT formation for the two strains. The data represent three biological replicates (a total of 12 cells for wt and 11 cells for Δ*sigD* strain were analyzed) ± SEM.

of wt *B. subtilis* cells (Supplementary Fig. 5 and Supplementary Movie 2).

In the Δ*sigD* strain, SYTOX penetration displayed slower kinetics than in wt cells and the dying cells took longer to form NTs (Fig. 3b, c). Moreover, the frequency of NT formation in Δ*sigD* (at 90 min) was lower than in wt (at 25 min) (Supplementary Fig. 6a, b compared to Fig. 2c, $p < 0.001$, GLM, two-sided, $z = −10$, 95% CI OR = 0.10–0.22). This validated our initial genetic screen based on SEM (Supplementary Fig. 1l). We further analyzed the Δ*lytEF* strain and found that it also formed NTs less frequently than wt (Supplementary Fig. 6c–e compared to Fig. 2c, $p < 0.001$, GLM, two-sided, $z = −9$, 95% CI OR = 0.01–0.04).

The signal arising from the Nile Red membrane stain that appeared in the newly formed NTs (ca. 25%) quantitatively corresponded with the decrease in the signal coming from the cell envelope (Supplementary Fig. 7a, b). This correlated with the amount of the newly created NT surface (Supplementary Fig. 7c), implying that the cell membrane was rapidly and aggressively utilized for NT formation.

We also asked, whether cells killed by other means (antibiotics) but grown without pressure in frames (Gene frames, Thermo Scientific) containing LB agar covered with coverslips would form NTs. Supplementary Fig. 8a–c shows that even in the control experiment under these conditions in the absence of any antibiotic a small fraction of the cells died, although no NTs were detected. Using ampicillin, which inhibits a transpeptidase involved in bacterial cell wall biosynthesis[34], led to death of majority of the cells and induced NT formation from dying cells although not as frequently as the P-GLG method (Supplementary Fig. 9a–c, $p < 0.001$, GLM, two-sided, $z = −8$, 95% CI OR = 0.01–0.05). Interestingly, NTs now originated mostly from the cell sides, to a lesser degree from the septal regions and not at all from cell poles (Supplementary Fig. 10). We also tested chloramphenicol, which inhibits the peptidyl transferase activity of the ribosome[35]; even though chloramphenicol is bacteriostatic, the concentration used did result in some dying cells and we detected a single NT at 90 min (Supplementary Fig. 11a–c). A similar result was also obtained using the RNAP-targeting rifampicin[36,37] (Supplementary Fig. 12a–c).

**NTs do not transfer non-conjugative plasmid DNA**. Previously, protein transfer was claimed to occur through NTs[1]. Using SIM we, indeed, detected both membrane-targeted GFP and cytoplasmic ZsGreen in NTs (Supplementary Fig. 13).

So far, we had observed NT formation solely from dying *B. subtilis* cells. However, we could not exclude the possibility that occasionally NTs might be formed from living (though likely highly stressed) cells, and we had not been able to detect these presumably extremely rare events. The presence of ZsGreen in NTs indicated that if NTs did connect two cells, then they could function to transport various molecules, and this might be of biological relevance as previously proposed. The best way to test this was to use the reported non-conjugative plasmid transport as the readout for the transfer. The advantage of this approach over, e.g., monitoring the transport of proteins or mRNA or metabolites is its extreme sensitivity. Even a single plasmid transfer among millions of cells can be detected.

Initially, we used an already published protocol[9,38], using a non-conjugative plasmid, pCPP31-Y1[39], bearing a chloramphenicol resistance gene. We incubated donor (containing the plasmid) and recipient (bearing chromosomally encoded macrolides (MLS) resistance gene) strains for four hours in LB. We then plated serially diluted cultures on LB containing various antibiotic combinations. Figure 4a shows that bacterial growth on the combination of both antibiotics was observed only at the highest cell density, suggesting that the transfer might, indeed, occur, though rarely. Subsequent tests revealed that the resulting strain contained both antibiotic resistance genes, one chromosomally encoded and the other plasmid-borne.

Next, we wished to determine whether the bacterial growth on both antibiotics was due to DNA transfer through NTs or to DNA uptake by the competence system that allows a bacterial culture to bind and take up high-molecular-weight exogenous DNA. The master regulator of competence in *B. subtilis* is the ComK transcription factor[17]. We began with the same donor and recipient strains as in the previous experiment and made several genetic alterations to them: (i) Δ*sigD*; (ii) Δ*comK*; and (iii) P$_{hyperspank}$-*comK*—a strain overproducing ComK [strains (ii) and (iii) also form NTs, Supplementary Fig. 14a, b]. Finally, we also examined each of these strains with or without DNase I treatment to remove any DNA originating from lysed cells. Figure 4b shows that all transfer that was observed in the absence of DNase I for each strain was abolished when DNase I treatment was included. Furthermore, the Δ*comK* strain showed no transfer even without DNase I treatment. Conversely, ComK overproduction greatly increased plasmid transfer. In the Δ*sigD* genetic background, the plasmid transfer was decreased. In the genetic backgrounds (i)–(iii), the transfer was quantitatively correlated with the expression of the *comK* gene (Fig. 4c). Moreover, it was previously reported that the absence of the strictly SigD-dependent *hag* gene decreases competence[40]. Subsequent analysis then revealed that it was plasmid and not chromosomal DNA that was transferred under these experimental conditions (Supplementary Fig. 14c). We note that a previous study showed the ability of *B. subtilis* to be transformed in LB, albeit with low efficiency, and this efficiency increased when *comK* was overexpressed, similar to our results[41]. Taken together, we conclude that NTs are unlikely to serve as conduits for non-conjugative plasmid DNA transfer between *B. subtilis* cells.

**Formation of NTs in different bacterial species**. Finally, we investigated how other bacterial species form NTs. We used (i) the gram-positive *Bacillus megaterium*, an industrially important organism and a relative of *B. subtilis*, (ii) the gram-positive *Deinococcus radiodurans*, an extremophile capable of surviving high

doses of radiation[42], and (iii) the gram-negative model bacterium *E. coli*. Figure 5 shows that all these species formed NTs to various degrees under the P-GLG setup. The NTs from *B. megaterium* were most similar to those from *B. subtilis*. *D. radiodurans* extruded a high number of NTs from each cell. *E. coli* reacted mostly by forming blebs, or vesicles, and occasionally also NTs. In all cases, NTs were formed only from dying cells.

## Discussion

Our study indicates that the bacterial membranous tubular structures, often called nanotubes, are formed when the cells are dying, or after cell death, as a consequence of the compromised cell wall and (likely) excess internal pressure (Fig. 6). Importantly, these NTs do not serve as channels through which non-conjugative plasmids can be transported. Instead, they are a sign of disintegrating cells, and we believe it is debatable whether they have a physiological role.

Our initial experiments were aimed at defining the genetic requirements for NT formation. We identified the SigD regulon as important for this process and this correlated with the known involvement of the CORE element (*fliOPQR, flhBA*)[9]. The loss of the CORE element genes leads to inactivation of SigD by preventing export and proteolysis of FlgM, the anti-SigD factor[43]. Further, a mutant strain lacking the autolysins LytE and LytF, of which the latter is SigD-dependent, proved to be similarly deficient in NT formation. As LytE and LytF localize to cell poles and Δ*sigD* or Δ*lytEF* mutant cells remain in chains and do not separate into single cells, the blockage of the cell poles/absence of cell wall degradation by LytE and LytF might have caused the delay in the appearance of NTs. Moreover, it was very recently reported that a Δ*lytBC* mutant displays reduced NT production[38]. Both these genes also belong to the SigD regulon.

In our experiments, NTs were formed exclusively from dying cells. Cell death was monitored by SYTOX. In some instances, NTs that were detached from dying cells could be occasionally associated with living cells, creating thus a misleading illusion (Supplementary Fig. 15). When observed by time-lapse microscopy, NT formation was a rapid process, in the order of seconds, during which the NTs used the cell plasma membrane for their extrusion. Occasionally, NTs contained terminal structures (see also Supplementary Movie 3); these were also detected in a previous study[9], but not commented upon.

We used various types of stress to cause cell death (pressure, different antibiotics); regardless of the stress applied, NTs were always detected associated with dying cells, though their relative amounts varied. The greatest number of NTs was formed in samples prepared by P-GLG method where pressure was applied to the coverslip (Figs. 2 and 3 and Supplementary Movie 1). Mechanistic details of the process and identification of forces involved therein will be addressed by a future study.

How, then, are NTs formed? We propose that, as the cells are dying and start disintegrating, weak spots in the cell wall may serve as channels through which NTs are extruded to release the intracellular pressure[44]. This is again consistent with the reported requirement for the CORE element in NT formation. The same components that are required for flagella to be assembled and traverse the cell wall likely create weak spots in the cell envelope that may serve as channels through which NTs can be extruded. Typically, NTs emerge from the cell poles. The appearance of NTs from other parts of the cell (sides) in, e.g., ampicillin-treated cells might stem from region-nonspecific structural damage to the cell wall. The tubular nature of NTs may then depend on the lipid composition of the membrane. Cardiolipin in combination with the PmtA protein from the phytopathogenic bacterium *Agrobacterium tumefaciens* was previously reported to promote

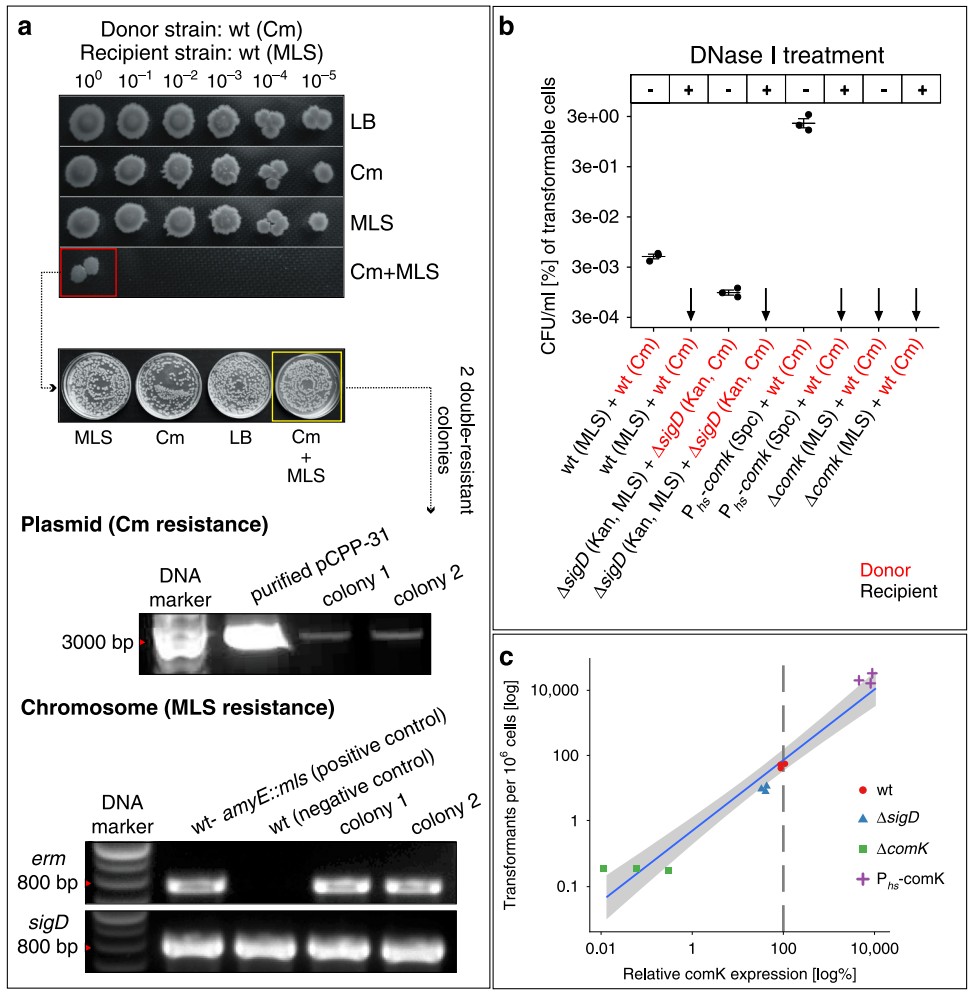

**Fig. 4 Non-conjugative plasmid DNA transfer. a** The donor (LK1925, wt containing the Cm resistance gene on plasmid pCPP31-Y1) and recipient (LK1922, wt harboring the MLS resistance gene on chromosomal DNA) strains were grown separately overnight. The following day, both strains were diluted to an initial $OD_{600} = 0.05$, mixed 1:1, and cultivated for 4 h at 37 °C. The mixture was then serially diluted into fresh LB medium and 2 μl were spotted on LB agar (control) and LB agar containing appropriate antibiotics (Cm-donor selection, MLS-recipient selection, Cm+MLS-cells with both antibiotic resistance genes). The cells were allowed to grow for 18 h at 37 °C. Only the mixture with the highest initial density then produced double-resistant cells. This colony was subsequently diluted into fresh LB medium to $OD_{600} = 0.05$ and 100 μl were plated on LB agar dishes with or without Cm, MLS or Cm + MLS. We detected many double-resistant colonies on the Cm + MLS agar dish (yellow square) and for two of them we confirmed the presence of the two resistance genes by plasmid isolation (the dominant supercoiled form of the plasmid is shown) and PCR of the chromosomally-located *erm* (MLS) resistance gene (primers against the *sigD* gene were used as a control). As the DNA marker we used GeneRuler DNA Ladder Mix (Thermo Scientific). All experiments were conducted in three biological replicates with similar results. **b** Donor (LK1925-wt or LK1944-Δ*sigD*) and recipient (LK1925-wt; LK2317-P_hs-*comK* [hs, hyperspank]; LK1940-Δ*sigD* or LK2380-Δ*comK*) strains were grown separately overnight, then diluted to an initial $OD_{600} = 0.05$, treated or not treated with DNase I, and mixed in a 1:1 ratio. The mixtures were inoculated into fresh LB medium (to an initial $OD_{600} = 0.05$) without antibiotics and gently shaken. After 4 h, bacteria were harvested and diluted to an $OD_{600} = 0.5$. 100 μl of each mixture was plated on LB agar dishes with appropriate antibiotic combinations, based on antibiotic resistance genes in donor/recipient. The combinations are specified in parenthesis after each strain below the graph. The dishes were incubated overnight at 37 °C and then CFU/ml were counted. Data are presented as mean values ± SEM, *n* = 3 biological replicates per condition shown as individual dots. Black arrows indicate zero values in all three biological replicates. All differences between DNAase(-) and DNase(+) conditions in wt, Δ*sigD* and P_hs-*comK* are statistically significant as well as all pairwise differences between the DNase(-) conditions (all *p* < 0.001, ANOVA adjusted with Tukey's HSD, two-sided). **c** Correlation between the number of transformants per 10⁶ cells and *comK* expression (determined by RT-qPCR). For display purposes we added 1 to all raw transformation counts. The blue solid line is the maximum likelihood linear fit, the shaded gray area represents the 95% confidence interval of the fit. The vertical dashed line indicates the wt expression against which the samples were normalized (100%).

formation of tubular structures in vitro[45,46]. Cardiolipin is an integral part of the *B. subtilis* membrane and might play a role in NT formation.

We have also noticed that the amounts of NTs (or filamentous, NT-like structures in the case of SEM) observed by SIM and/or SEM are sensitive to growth conditions (solid or liquid media) and the conditions used for preparation of the microscopic sample. Supplementary Figs. 16 and 17 illustrate this issue. Cells

grown in liquid media and visualized by SIM or SEM display few NTs (Supplementary Fig. 16a–d). Cells grown on solid media and visualized by SIM form no NTs under the conditions tested in our study (Supplementary Fig. 16e, f, h) and we assume that this is due to the minimal manipulation required for this imaging method. To the contrary, when we used two previously published sample preparation protocols for solid media-grown cells, we detected many NT-like structures by SEM (Supplementary

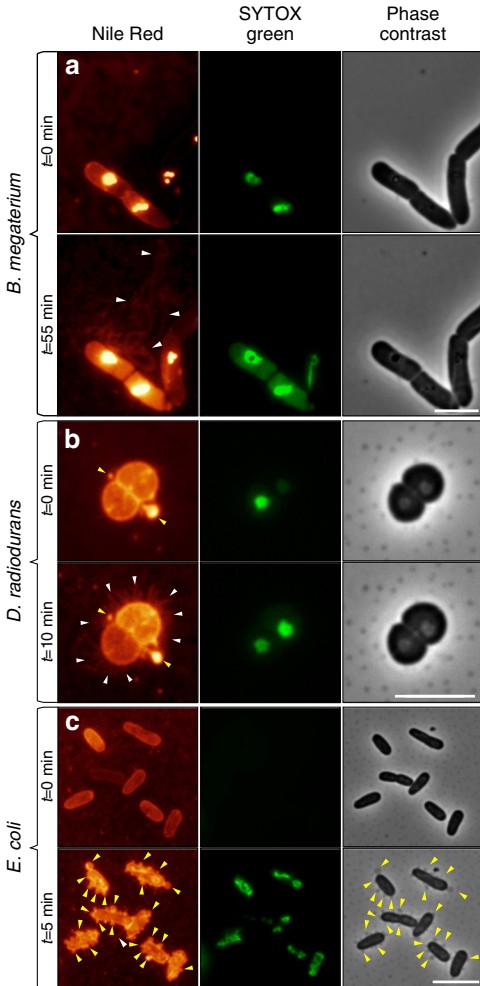

**Fig. 5 Formation of NTs from selected bacterial species. a** *Bacillus megaterium* (LK1780) was grown to exponential phase. The microscopic samples for this and other organisms shown in this Figure were prepared as in Fig. 3 (the P-GLG method). After 55 min under the coverslip, the cells formed NTs that were similar to those of *B. subtilis*. **b** Exponentially growing cells from the extremophile bacterium *Deinococcus radiodurans* (LK1553). **c** Exponentially growing cells of gram-negative *E. coli* (LK1133). In all samples, membranes were stained with Nile Red (false colored glow) and SYTOX green (green); the third column shows phase-contrast images. All experiments were performed three times. Yellow arrows indicate membranous blebs and vesicles, white arrows indicate NTs. Scale bar = 5 μm.

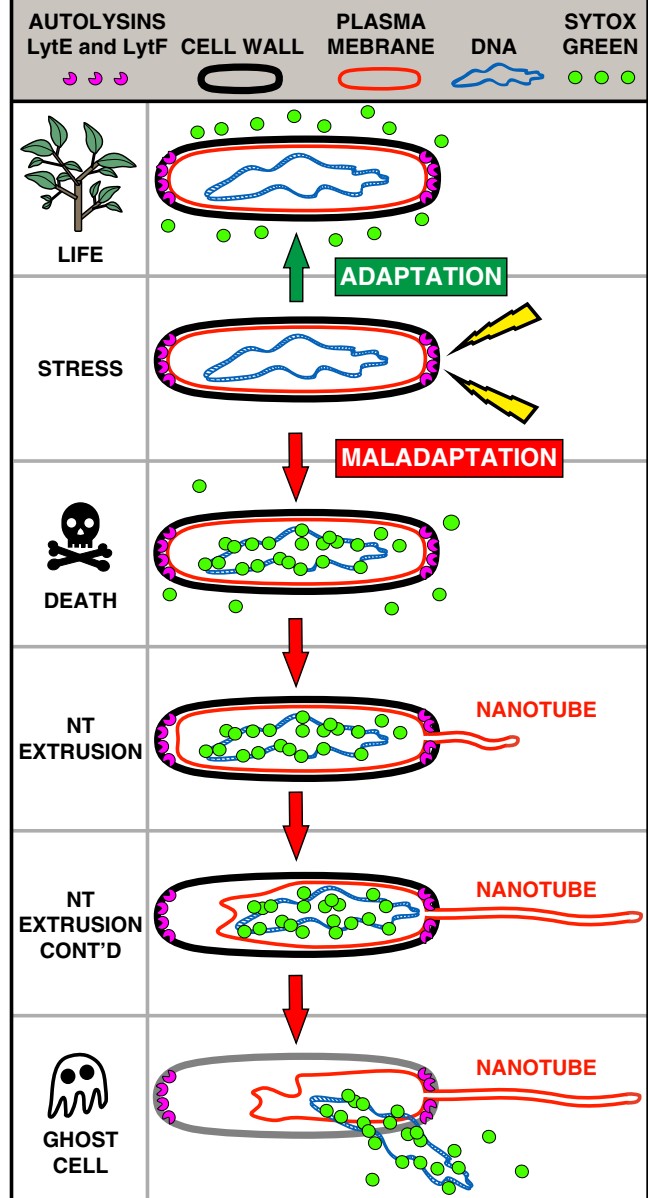

**Fig. 6 Model of NT formation.** The figure depicts NT formation: stressed cells (e.g., pressure, ampicillin treatment) either adapt or when the stress is too severe, die. During cell death, the plasma membrane becomes compromised and membranous tubular structures are extruded from dying cells. Autolysins LytE and LytF that localize at the cell poles weaken the cell wall and facilitate NT extrusion. Eventually, the cell contents are emptied and a hollow husk—the ghost cell—remains. No physiological role for these NTs has been detected.

Figs. 16e, g, h and 17c–f). What do these NT-like structures consist of? Apparently, they are not made of membranes as shown in the SIM images (Supplementary Figs. 16b and 17b). We speculate that they might be derived from the disrupted extracellular matrix. The disruption is likely caused by imprinting of the bacteria on EM grids and/or subsequent manipulations with the grids (Supplementary Fig. 17c–f). Consistently, cells grown on cellophane positioned on LB agar and imaged by SEM without the requirement for EM grid imprinting display undisrupted extracellular matrix (Supplementary Fig. 17a). To summarize this part: the more gentle the conditions (Supplementary Fig. 17a, b), the fewer (if any) filaments detected.

A most pressing question then is whether NTs in fact have any physiological roles. Although we did not detect any NTs associated with live cells by microscopy, we screened for their potential to serve as channels for transferring non-conjugative plasmids. Our screen revealed that this transfer was exclusively dependent on the cell ability to take up exogenous DNA, thereby

ruling out any NT involvement. Moreover, although NTs were reported to connect cells and in some of our images we detected such cell pairs, we believe that these connecting NTs are artifacts of the microscopy techniques—the NTs likely emanate from one cell and their entanglement with other, nearby cells creates the illusion of a connection. In any case, the occurrence of NTs is rare and these connecting NTs are even less frequent. Furthermore, NTs were previously reported to transport metabolites such as amino acids and this phenomenon was contact (i.e., NT)-dependent[2,14]. In light of the evidence presented here, it seems more likely that the metabolites taken up by the auxotrophic cells in these studies were released from nearby lysed cells.

In addition to *B. subtilis*, we also observed membranous tubular structures emerging from dying cells of other bacterial species (*B. megaterium*, *D. radiodurans*, *E. coli*). The literature contains several reports of membranous tubular structures emerging from still other bacterial species; interestingly, some type of stress typically induces these structures. An example is *Myxococcus xanthus* where these structures are termed outer membrane tubes (OMTs). OMTs were found to contain outer membrane (OM) proteins and lipids but no other cytoplasmic material. Massive OMT formation induced by stress (lack of oxygen; addition of metabolic inhibitors) then blocks the intercellular transfer of OM proteins, suggesting that the OM protein transfer between cells is not mediated by OMTs. Rather, the transfer depends on the TraAB system and direct cell-to-cell contact[8,47,48]. Furthermore, Wei et al.[8] published phase-contrast images, suggesting that it is predominantly ghost cells that are associated with OMTs. Although these authors discussed that OMT formation was caused by cell stress they did not explore the possibility that some cells were dying[8]. Finally, OMTs were also found in the pathogens *Francisella novidica* and *Francisella tularensis* where OMTs were stimulated by stress conditions induced by amino-acid deprivation or during macrophage infection[49,50]. Nevertheless, these structures, being formed from the outer membrane, do not have to cross the cell wall and, therefore, the mechanism of their formation is likely different from that one of *B. subtilis* NTs. Finally, NT-like structures were formed by *D. radiodurans* when challenged with mitomycin C[51].

Various types of tubular structures have also been reported for eukaryotic cells and mitochondria. In some cases, the formation of these structures was induced by the presence of bacteria. Tunneling nanotubes (TNTs) of macrophages are an example[52,53]. TNTs are long-range membranous F-actin-containing tubes that are classified into two types based on their thickness and the presence or absence of microtubules[54]. TNTs, e.g., help spread the HIV-1 virus and their formation is stimulated by coinfection with *Mycobacterium tuberculosis*[55]. TNTs, however, appear to be distinct from bacterial NTs by the presence of a proteinaceous scaffold and appear to be bona fide channels for cell-to-cell communication.

We conclude that *B. subtilis* nanotubes are a hallmark of dying cells, or cell death, and are involved in the final cell disintegration. In other bacterial species, similar structures should be studied with utmost care before attributing physiological roles to them.

## Methods

**Media and growth conditions**. *B. subtilis*, *B. megaterium*, and *E. coli* strains were grown at 37 °C in Luria-Bertani (LB) media, supplemented, when needed, with ampicillin (Amp, 100 μg/ml), spectinomycin (Spc, 100 μg/ml), chloramphenicol (Cm, 5 μg/ml), tetracycline (Tet, 10 μg/ml), kanamycin (Kan, 10 μg/ml), and MLS (lincomycin 25 μg/ml and erythromycin 1 μg/ml). One milimolar IPTG (isopropyl-D-thiogalactopyranoside, Amresco) was added where indicated. *D. radiodurans* was grown in TGY medium (1% Bacto Tryptone; 0.5% yeast extract; 0.1% glucose).

**Bacterial strain construction**. Bacterial strains are listed in Supplementary Table 1, and primers used for the validation of gene deletions are listed in Supplementary Table 2. Genomic DNAs (gDNAs) were isolated using the High Pure PCR Template Preparation Kit (Roche) and PCR reactions were carried out using an Expand High Fidelity PCR System (Roche).

**SEM procedure**. Sample preparation and processing were carried out essentially as in[56] but with some modifications. In brief, exponential cultures of *B. subtilis* strains (OD$_{600}$ ~ 0.6) were pre-fixed with 3% glutaraldehyde in culturing media at room temperature (RT), washed with cacodylate buffer and fixed with 3% glutaraldehyde in cacodylate buffer at 4 °C overnight. These extensively washed cells were then sedimented onto poly-L-lysine-treated circular glass coverslips at 4 °C in a Petri-dish moist chamber for 42 h. The cells attached to the coverslips were washed three times and postfixed in 1% OsO$_4$ for one hour at room temperature and again washed three times. The coverslips were dehydrated in a graded ethanol series (25, 50, 75, 90, 96 100, and 100%) followed by absolute acetone and critical point dried

in a K850 Critical Point Dryer (Quorum Technologies Ltd, Ringmer, UK). The dried samples were sputter-coated with 3 nm of platinum in a Q150T Turbo-Pumped Sputter Coater (Quorum Technologies Ltd, Ringmer, UK). The final samples were examined in a FEI Nova NanoSEM scanning electron microscope (FEI, Brno, Czech Republic) at 3 or 5 kV using ETD, CBS, and TLD detectors and using SEM software Helios NanoLab. The beam deceleration mode was used when sample charging occurred[57].

**SIM procedure**. For membrane staining, Nile Red was added to 1 ml of exponentially growing culture at a final concentration of 10 μg/ml. After 10 min of incubation at RT, bacteria were pelleted and washed once with 1× PBS. For staining of flagella, Alexa Fluor 488 maleimide conjugate (5 μg/ml, final concentration) was added and samples were washed twice with 1× PBS, then stained with Nile Red. The bacteria were subsequently re-suspended in 1× PBS, spotted on a coverslip covered with a thin agarose pad (1.5% agarose in 1× PBS supplemented with 10× diluted LB medium). Alternatively, for pressure experiments, samples were spotted on poly-L-lysine glass slides and covered with a poly-L-lysine coated coverslip. The samples were observed using a DeltaVision OMX™ equipped with a 60× 1.42, PlanApo N, oil immersion objective and softWoRx™ Imaging Workstation software. ZsGreen and GFP-tagged protein or Alexa Fluor maleimide conjugate were imaged using 488 nm excitation; Nile Red was imaged using 568 nm excitation.

**Time-lapse fluorescence microscopy**. To visualize the membrane structures, 1 ml of exponentially growing bacterial culture was stained by Nile Red (10 μg/ml) or by FM4–64 (1 μg/ml), washed twice with 1× PBS and re-suspended in 1× PBS containing 1 μM SYTOX green. The sample was then immediately either spotted onto a poly-L-lysine glass slide and covered with a poly-L-lysine coverslip (GLG method or P-GLG method when pressure ~80 kPa was applied for 10 s) or spotted on a 1× PBS agarose pad covered with a non-coated coverslip (GAG method). Pictures were obtained at the indicated time points (*t* = 0 is the start of microscopy, typically 30 s after coverslip addition) using a Olympus BX63 fluorescence microscope equipped with a Andor Zyla 5.5 sCMOS camera (alternatively, an Olympus IX81 microscope equipped with Hamamatsu Orca/ER camera was also used). Olympus CellP imaging software or Olympus Image-Pro Plus 6.0 software was used for image acquisition and analysis.

**MS analysis of cytoplasmic membrane fractions**. Cytoplasmic membrane fractions from the *B. subtilis* wt (LK1432) and Δ*sigD* (LK1873) strains were prepared following a previously described protocol with some modifications[11]. Briefly, cells were gently shaken and grown until they reached exponential phase (OD$_{600}$ = 0.6) and then pelleted (6000 × *g*, 10 min at 4 °C). Samples were re-suspended in 20 ml of 1× P buffer (35% NaCl; 35% Na$_2$HPO$_4$; 10% glycerol; v/v) containing 3 mM 2-mercaptoethanol and 1 mM serine protease inhibitor PMSF (phenylmethylsulfonyl fluoride) and sonicated on ice for 10× 10 s, with probe amplitude 0.5, (Hielscher sonicator, UP 200 s). The lysates were then centrifuged (6000 × *g*, 20 min at 4 °C), the supernatants were transferred to clean ultracentrifuge tubes, and the membranes were collected by centrifugation at 100,000 × *g* for 45 min at 4 °C. The supernatants were subsequently discarded and the pellets containing the membrane fractions were dissolved in 1× P buffer and analyzed by MS.

The proteins were analyzed by trypsin filter digestion in the form of an eFASP (enhanced Filter-aided sample preparation) method[58]. Briefly, the samples were reduced, alkylated and digested by trypsin on YM-10 Microcon filters (Merck). The resulting peptides were then desalted on a C18 SPE column (PepClean, Thermo). The peptides were separated during LC-MS/MS using a nano-LC system (Ultimate 3000 RSLC nano, Dionex) on an Acclaim PepMap C18 column (75 um Internal Diameter, 250 mm length) by applying a 125 min acetonitrile elution gradient in 0.1% formic acid. The chromatographic column was connected via nanoESI to a tandem mass spectrometer (TripleTOF 5600, Sciex). We collected data for the identification and the quantification of the proteins in one measurement sequence. Employing two methods using the same chromatography parameters but different mass spectrometric data collection setups [the first: the data-dependent analysis (DDA) method; the second: the data independent method (DIA—SWATH[59]], we measured the samples in a sequence of two consecutive runs of each sample (the first run aimed at collecting the DDA data to identify the proteins; the second aimed at collecting the DIA data for to quantify them). All resulting DDA spectra were searched together using Protein Pilot 4.5 (Sciex) against the Uniprot *B. subtilis* reference protein database (downloaded 8th of October 2015), thus creating the library used for SWATH processing in PeakView 2.2 (Sciex). In the SWATH analysis the retention time was aligned in all the samples by selecting the peptides common across the retention time range for all the samples. For quantification purposes, we allowed up to 30 peptides per protein, 6 transitions per peptide, a peptide confidence of 95% and a false-discovery rate threshold of 1%. The processing steps produced an intensity of transitions, peptides and proteins. The final protein table was processed in MarkerView (Sciex) in order to create a statistical evaluation. The Student's *t*-test was performed on the monitored groups.

**Spotting assay—non-conjugative plasmid transfer**. Donor [LK1925, wt containing the Cm resistance gene on plasmid pCPP31-Y1)[39]] and recipient (LK1922, wt harboring the MLS resistance gene on chromosomal DNA) strains were grown

separately overnight. Next day, the strains were mixed (equal numbers of cells), diluted to an initial $OD_{600} = 0.05$ and cultivated for 4 h at 37 °C with shaking. The mixtures, serially diluted, were spotted on LB agar (without antibiotics) and LB agar containing appropriate antibiotics (Cm for donor selection, MLS for recipient selection, Cm+MLS for recipients that acquired the non-conjugative plasmid). After 18 h, the dishes were photographed and double-resistant colonies were inoculated into LB to $OD_{600} = 0.05$. 100 μl of bacterial suspensions were plated on LB agar with or without antibiotics (the same as above). Double-resistant colonies were inoculated into LB medium with antibiotics (Cm + MLS) and grown overnight at 37 °C. On the following day, their genomic (High Pure PCR Template Preparation Kit) and plasmid (see: *B. subtilis* plasmid isolation) DNAs were extracted. The presence of the *erm* gene (MLS$^r$) in double-resistant strains was verified by PCR (Expand High Fidelity PCR System).

**B. subtilis plasmid isolation**. Two milliliter of overnight culture was pelleted (5 min, $13,200 \times g$ at 4 °C) and washed twice with 500 μl of TES (0.02 M, Tris-HCl pH 8; 5 mM, EDTA; 0.1 M, NaCl). The supernatant was discarded and 40 μl of lysozyme-containing buffer (30 mM, Tris-HCl pH 8; 50 mM, EDTA; 50 mM, NaCl; 25%, sucrose; lysozyme 500 μg/ml) was added. After 15 min at 37 °C, the sample was placed on ice. 160 μl of SDS buffer (2 ml of 10% SDS; 2 ml of 0.5 M EDTA pH 8; 12 ml of TES buffer) was mixed with 50 μl of 5 M NaCl and added to the sample, gently mixed and left on ice for 60 min. The mixture was subsequently centrifuged (30 min, $13,200 \times g$ at 4 °C) and the supernatant containing the plasmid DNA was transferred to a clean micro centrifuge tube. Finally, the plasmid DNA was precipitated with ethanol and analyzed on an agarose gel.

**Non-conjugative plasmid transfer**. Donor (harboring plasmid pCPP31-Y1, Cm$^r$) and recipient (containing MLS$^r$ in genome) strains were grown separately overnight. The next day, cultures were pelleted ($9000 \times g$, 3 min at RT) and dissolved in 1× DNAse I buffer diluted in 1× PBS. DNAse I was then added (or not for the negative control) and samples were incubated for 20 min at 37 °C. Donor and recipient strains were mixed (or each strain was grown separately) at a 1:1 ratio at an initial $OD_{600} = 0.05$ into fresh LB medium without antibiotics. Cultures were gently shaken for 4 h at 37 °C. Subsequently, bacteria were diluted to $OD_{600} = 0.5$ and 100 μl were plated on LB agar dishes with appropriate antibiotics (MLS-recipient selection; Cm + MLS—selection for double-resistant bacteria). CFU/ml were counted on the following day. The efficiency of plasmid receipt by recipient cells was expressed as the ratio of double-resistant cells (those able to grow on Cm + MLS) to all recipient cells in the mixture that grew in the presence of MLS.

**Quantitative PCR**. Two milliliters of exponentially growing cells (LB medium with 200 μM IPTG, gentle shaking) [wt (LK1432), Δ*sigD* (LK1873), Δ*comK* (LK2380), P$_{hs}$-*comK* (LK2317); $OD_{600}$ ~0.8] were treated with RNAprotect Bacteria reagent (QIAGEN), pelleted and immediately frozen. Their total RNA was isolated with the RNeasy Mini Kit (QIAGEN). Prior to RNA extraction, recovery marker RNA was added [a fragment of 16 s rRNA from *M. smegmatis* (amplified by primers nos. LK1281 and LK1282, see Supplementary Table 2)]. Finally, RNA was DNase treated (TURBO DNA-free™ Kit, Invitrogen). Five micrograms of total RNA was reverse transcribed to complementary DNA with reverse transcriptase using random hexamers as primers (SuperScript™ III Reverse Transcriptase, Invitrogen). This was followed by qPCR in a LightCycler 480 System (Roche Applied Science) containing LightCycler® 480 SYBR Green I Master and 0.5 μM of each primer. The primers were designed with Primer3 software and their sequences are given in Supplementary Table 2. The data were normalized to the recovery marker and the number of cells.

**Growth of B. subtilis in the presence of antibiotics**. One milliliter of *B. subtilis* exponential culture was stained with Nile Red (final concentration 10 μg/ml) for 10 min at RT and washed once with 1× PBS. LB agarose containing 1 μM SYTOX green and an appropriate antibiotic (Amp, 500 μg/ml; Cm, 5 μg/ml; Rif, 50 μg/ml) was prepared inside a gene frame (Invitrogen). For time-lapse experiments, it was necessary to create a narrow agarose strip (~2.5 mm) in the middle of the gene frame (by removing agar from its sides) to allow oxygen to efficiently diffuse through the sample. The Nile Red-stained culture was then spotted on agarose and covered with a coverslip that was held by the gene frame (no pressure) and placed in a chamber at a constant 37 °C temperature. Pictures were taken with an Olympus CellR IX81 detection and analysis system equipped with a Plan-Apochromat 100×/1.45 NA oil objective and an EMCCD Hamamatsu camera.

**Diverse methods for high-resolution bacterial culture observation**. Exponential bacterial culture ($OD_{600} = 0.6$) was diluted to low density ($OD_{600} = 0.05$) and spotted on cellophane lying on an LB agar dish and grown for 4 h. In all, 4 × 2 cm cellophane strips containing bacteria were carefully taken and mounted onto glass slides using Scotch tape. The slides were placed into a glass desiccator with a small container of 2% OsO$_4$ in double-distilled water (ddH$_2$O) and the cells were allowed to fix at room temperature for several days[60]. The fixed cellophane strips were then cut into pieces in size of standard SEM mounts (12.5 mm) and mounted with

conductive tape onto aluminum mounts (SPI). The samples were then sputter-coated with 3 nm of platinum. The SEM analysis was essentially done as described above.

As an alternative to the cellophane SEM experiment, we used a SIM-based approach. Here, instead of cellophane, a thin ~1 mm 1× PBS agarose pad was used. The pad was positioned on an LB agar dish containing 10 μg/ml Nile Red. After 4 h, the agarose pad was removed, covered with a coverslip and observed by SIM (see: SIM procedure).

Subsequently, two previously published protocols for observing NTs were used with minor changes[1,9,11]. First, bacterial cells were grown directly on LB agar dishes for 4 h at 37 °C and then imprinted onto glow-discharge activated EM grids[61]. The grids were fixed with drops of 2.5% glutaraldehyde in sodium cacodylate buffer in Petri-dish for 20 min[1]. The fixed grids were washed three times with ddH$_2$O and dehydrated in a graded ethanol series (25, 50, 75, 90, 96, 100, and 100%), and air-dried directly from 100% ethanol. Finally, the grids were sputter-coated with 3 nm of platinum and mounted into a transmission electron microscopy grid table for SEM examination. A FEI Nova NanoSEM scanning electron microscope (FEI, Brno, Czech Republic) at 5 kV using ETD, CBS, and TLD detectors in beam deceleration mode was used for SEM analysis.

Second, exponentially growing cells were spotted at $OD_{600} = 0.05$ onto glow-discharge activated formvar/carbon film coated EM grids placed on a nitrocellulose membrane[11]. These grids were taken after 4 h of cultivation (at 37 °C) on an LB agar dish and fixed with drops of 3% glutaraldehyde in a petri-dish most chamber. The fixed grids were subsequently processed as described above.

**Image analysis**. The final adjustment of the fluorescent images was done using Fiji ImageJ or the Analysis 3.2 Pro software suite (Sis GmbH; Olympus, now EMSIS GmbH) for SEM images. The free software Gimp (https://www.gimp.org/) and Inkscape (https://inkscape.org/) were also used for final image plate setup.

**NTs vs. cell membrane signal calculation**. Measurement of the membrane signal in the time-lapse images was done following bleaching correction. NT and cell membrane signals were measured at each time point. The background value was subtracted from the NT and cell membrane signal, yielding the final signal intensity.

**SYTOX green penetration**. The SYTOX green signal was measured at each time point in the time-lapse images. The background was subtracted and the highest value was normalized to 1. Combination with the membrane staining signal allowed the NT formation time point to be detected.

**Quantification and statistical analysis**. For each experiment we had at least three biological replicates. Averages of individual cells from different replicas are reported. The number of analyzed cells is given in the charts and figures. NT quantification was done manually. MS Excel and SigmaPlot were used for all statistical analyses, data processing, and presentation. For testing uniformity of distribution of NTs over cell we assumed the auxiliary hypothesis that poles cover <50% of cell surface. $p$-value for this joint hypothesis was computed using quantiles of the Beta distribution. Comparison of cell counts was performed with a binomial generalized linear model using the `glm` function in R[62] and—as an additional check—the `stan_glm` function from the RStanArm package[63]. Analysis of plasmid transfer and relative expression was performed on the log scale using a linear model with the `lm` and `cor.test` functions in R. Where applicable, contrasts and multiple testing corrections were performed with the `TukeyHSD` function. The difference of spatial distributions of NTs between wild type and Ampicilin-treated bacteria was assessed with Chi-squared test using the `chisq.test` function in R. Where applicable, we used models with full interactions. Whenever multiple model formulations were considered, we have reported the $p$-value least favorable to our conclusions. See "Code availability" for details regarding code for the statistical analysis.

**Reporting summary**. Further information on research design is available in the Nature Research Reporting Summary linked to this article.

## Data availability
Original microscopic images are available upon request. All data generated or analyzed during this study are included in this published article and its supplementary files, or available upon request. Source data are provided with this paper.

## Code availability
Full code for the statistical analysis can be found at Zenodo (https://doi.org/10.5281/zenodo.3999744).

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

## Acknowledgements

This work was supported by Czech Science Foundation-Grant No. 19-12956S (to L.K.), Ministry of Education, Youth, and Sports of the Czech Republic-project LO1509 (to O.K. and O.B.) and ELIXIR_CZ project LM2018131 (to M.M.), VEGA–Grant No. 2/0007/17 from the Slovak Academy of Sciences (to I.B.), a Grant from the Slovak Research and Development Agency under contract APVV-14-0181 and APVV-18-0104 (to I.B.), Czech Ministry of Health (Grant No. 17-29680A to L.K.). We thank C. Condon and J. Bauer for critically reading the manuscript.

## Author contributions

J.P., L.K., I.B. conceptualized the project. J.P., I.B., K.M., D.V., M.H., O.B., O.K., M.Š., H.Š. performed the experiments. M.M. performed statistical analyses. L.K., J.P., I.B., and O.B. wrote the first draft of the manuscript. All authors read and contributed to the final version.

## Competing interests

The authors declare no competing interests.
