## [Peer Review File · Nature Communications]

REVIEWER COMMENTS

Reviewer #1 (Remarks to the Author):

In their manuscript, Pospíšil and co-workers analyse the formation of bacterial structures termed "nanotubes". These poorly understood and controversial cell envelope-associated structures forming cell-to-cell contacts are implicated in inter- and cross species nutrient and DNA exchange. While these structures have been observed by few groups independently, there have also been substantial difficulties in reproducing the observations by many other groups. In this paper, Pospíšil and co-workers provide strong evidence that the formation of the nanotube-structures are highly dependent on the experimental conditions used, thus providing the first coherent explanation why the reproduction of the nanotube structures has been challenging. Furthermore, the authors provide compelling evidence that the formation of nanotubes is associated with the cell lysis process, and that the observed exchange of DNA is in fact facilitated by the *B. subtilis* competence machinery, rather than via nanotubes. Both these findings severely question the postulated models linking the observed nanotube-structures to specific biological functions. The study is highly timely and will be of considerable interest for those working on bacterial cell envelopes. While this manuscript will certainly not remain the "final word", it is a highly important contribution to a controversial topic. I therefore strongly endorse the publication of this manuscript in Nature Communications, subject to minor corrections listed below.

Minor concerns and corrections:

Lines 116-117:

Since the observation that cells producing nanotubes are losing phase contrast is quite central for this manuscript, a more thorough explanation of what "phase contrast" means is needed. I don't know what the authors mean with "loss of granularity". Please include an appropriate explanation how phase contrast relies on refractive index differences between cell interior and extracellular medium, loss of which indicating substantial leakage of cellular content.

Line 118 and throughout the manuscript:

While the authors rightly link the loss of phase contrast to cell death, the use of wording with respect to death and lysis is somewhat confusing throughout the manuscript. Both loss of phase contrast and cytoplasmic Sytox Green/Propidium iodide signals are perfectly valid indicators for cell lysis and, consequently, likely predictors for cell death. However, at which stage of the cell lysis process the cell in fact loses viability is unknown. In contrast, cells can very well lose viability without losing membrane integrity or phase contrast. Therefore, authors should appropriately explain the correlation with these two processes and, throughout the manuscript, refer to the cells losing phase contrast/gaining Sytox Green signal as cell undergoing lysis, rather than oversimplistically as "dead cells".

Line 139:

Sytox Green staining is not an indicator for impaired membrane potential. Rather, it is unable to diffuse across intact membranes and only stains cells with cells with large pores formed by antibiotic or proteins, or when the cell undergoes lysis. Cells with low membrane potential do not, per se, become permeable for Sytox Green. Please correct appropriately.

Figure 1 (and throughout the manuscript):

It is not clear what the authors mean with T=0 min. Assembling the slide, finding the cells under the microscope and imaging always take "some time" making 0 min certainly an incorrect value. Please provide a realistic estimate in the respective legends what the T=0 min represents in real terms (time from coverslip addition to image capture).

Figure 2A:

Please provide and overlay between the Nile Red and Sytox Green images. Replace life/dead cells with intact/lysed cells.

Line 144-145:

Poly-lysine penetrates bacterial membranes resulting in growth inhibition and loss of membrane potential (Katsu et al, Biochem Biophys Res Commun, 1984) making it perhaps not so surprising that poly-P aggravates the cell lysis process. Please mention the antibacterial, membrane-acting properties of poly-lysine is an appropriate place of the manuscript.

Line 157:

It is not clear here if the authors mean nanotubes emerging within seconds of coverslip/pressure application, within second of Sytox Green entry, or something else. Please rephrase for clarity.

Figure 3A:

Please provide quantification of the cell Sytox Green signals, and the cell phase contrast signals (for instance as the delta of the average pixel inside vs outside of the cell) to accompany the individual microscopy images shown in panel A and B. This is to show if there really is a sudden increase in Sytox Green signal and phase contrast associated with the emergence of the nanotubes. This is not really clear from the images themselves.

Lines 179-180 and Figure S7:

-Please include the phase contrast images for the cells shown in panels S7A-D.

-The membrane patches shown to emerge in Chloramphenicol and Rifampicin-treated cells (highlighted with arrows) look very much distinct from those observed with Ampicillin or Poly-P. I am not convinced such intracellular or membrane-associated patches can by any means be interpreted as nanotubes. Please re-analyse and strictly limit the quantification to structures extending away from the cell surface only. This will likely also need some toning down the statement in lines 179-182.

Lines 244-248:

-As in many other place in the manuscript, please do not refer to these cells as "dead". While Sytox Green is commercially sold as part of "life-dead staining", this more a sales pitch than a valid scientific statement. It is strictly a reporter for membrane integrity, not cell viability.

-I am not convinced by the author's interpretation that increased Nile Red staining indicates an ability to cross the membrane and stain the "inner leaflet". Nile Red is commonly used to stain intracellular fatty acid droplets in eukaryotes, and is perfectly able to stain Bacillus endospore membranes while still encased by the mother cell membranes. Both are strong indications that Nile Red can diffuse across intact membranes. I would omit this likely wrong and unnecessary speculation altogether.

Lines 267-268:

-The curvature dependency of Cardiolipin described in reference 39 (Beltran-Heredia et al.) refers to membrane curvatures order of a magnitude (and more) stronger than those observed at bacterial cell poles. While such a mechanism could well be involved in the extrusion of the thin, tubular nanotube-structures, I cannot follow the logic how this could explain the preferential emergence from cell poles (especially given the localisation of Cardiolipin at B. subtilis cell poles has recently been questioned (Pogmore et al, 2018). Please rephrase.

-Also, lipids don't "bind" curved membranes but localise to/enrich at curved membrane areas.

Lines 292-302:

Given the very different composition, architecture and the cellular context, it is exceedingly likely that the mechanisms governing the formation of outer membrane tubes and inner membrane

"nanotubes" are distinct. Therefore, discussing the observations of this paper in the context of OMTs seems a bit speculative. This is especially given it is unlikely that an outer membrane is affected from cell wall-linked lysis the same way as the inner membrane is. Please provide a more nuanced discussion here.

Reviewer #2 (Remarks to the Author):

The manuscript by Prospisil et al critically examines a striking and enigmatic phenomenon whereby bacteria, most notably *Bacillus subtilis*, form long narrow membrane filaments by an unknown mechanism. These structures, called nanotubes (NTs), received considerable recognition since a landmark publication almost a decade ago (ref 1). That publication raised skepticism in the field about claims that NTs indiscriminately form cell-cell conduits between cells of the same species and distantly related species wherein macromolecular content in the cytoplasm is exchanged. Those and other concerns were partly addressed in subsequent publications (e.g. refs 9, 11 and 15). However, a lingering concern was that a single lab did all of the work and that an independent lab had not corroborated the rather revolutionary claims (refs 1, 9, 15, 16). Therefore, this work by Prospisil et al, is important because they are the first outside group to examine *Bs* NTs. These authors convincingly show that *Bs* NT production is caused by cell stress/death leading to punctures in the cell wall and the cytoplasmic membrane being extruded as NTs. These findings were shown by a variety of fluorescent and EM techniques using different reporters and mutant strains. They also demonstrate that cell-cell transfer of cytoplasmic plasmids are not mediated by NTs, as previously claimed, but instead occurs by donor cell lyses and DNA uptake by natural competence in the recipient cell. This work is carefully done, well written and the findings will have a broad impact to the microbiology field and will stimulate future research.

Specific comments:

Line 35: The first paragraph is a bit murky as different membrane tube structures are discussed as if they are the same, which they are not. Possible solution, change "These structures may serve different functions in these different organisms" to "These structures are composed of cytoplasmic or outer membranes, depending on the species of origin, and may serve different functions in these organisms."

Line 44: For clarity, please change "types" to "configurations"

How do the authors know the narrow (red arrow) filaments in Sup. Fig. 1 SEMs (and elsewhere) are not exopolysaccharide (EPS) bundles or filaments.

Line 88-9: Please discuss why the frequency of NTs is dramatically lower (1/5,000 cells) compared to prior reports.

Prior studies found that NTs were primarily made by cells in biofilms. Here the investigators used planktonic cells. Are there more NTs when cells are grown as biofilms?

The authors should incorporate a very recent publication into their discussion (e.g. line 129): Baidya, Rosenshine & Ben-Yehuda. Donor-delivered cell wall hydrolases facilitate nanotube penetration into recipient bacteria. 2020, Nat Comm.

Line 140: SYTOX is stated twice in one sentence.

The experiments in Figure 4 nicely and critically show that cell-cell transfer occurs via naked DNA and natural competence; not mediated by NTs. However, the transfer assay was done in liquid. In

prior studies, e.g. ref 1, transfer was done when cell were spotted on agar (biofilm-like). What happens if the experiments are instead done on agar? This is relevant because DNA transfer mechanisms could be fundamentally different between liquid and solid media.

Fig. 4B legend needs clarification in context of "appropriate antibiotics." It seems in every case at least two antibiotics were added and perhaps in some cases three?

The legend to Supp Fig. 5a is not clear. The bar graph seems to show percentage of cells with NTs, but, for example, what is the difference between "Live cells" and "Live cells NTs?" Also, in panel B, why are the cells bright green (SYTOX positive) at $t = 0$?

Supp Fig. 7/legend. No "white arrows" are shown; only yellow arrows. Please correct. In the bar graphs what does "Percentage" represent, e.g. in "Live cells NT"? It sounds like percentage of cells with NTs that have NTs.

Supp Fig. 9. Add white arrow(s) to panel C.

Line 297-99: These sentences are not accurate. Change "...phase-contrast images showing that it is predominantly 'ghost' cells that form OMTs" to "phase-contrast images suggest 'ghost' cells could contribute to the formation of OMTs." Also change "The authors did not discuss this phenomenon but those cells were most likely dead." to "Although these authors discussed OMT formation was caused by cell stress they did not emphasize that some cells might be dead."

Figure 6 nicely summarizes the findings in the paper.

Reviewer #3 (Remarks to the Author):

Here, Pospíšil et al investigate membranous nanotubes previously observed connecting bacterial cells. While previous work has hypothesised functional roles for these structures, Pospíšil et al conclude that these are manifestations of cell death, formed from purely biophysical forces and are of no functional significance. If that conclusion is correct, then the result is indeed interesting and I would recommend acceptance in Nature Communications. That said, there is a major concern with the validity of the results because no statistical analysis is ever presented. If the statements made were backed up by properly calculated stats and p-values I would recommend publication. See below for details and other major concerns. Until those are in place, it's basically impossible to adequately review the manuscript.

The major issue is that there are no statistics. No p-values are ever shown for any of the statements made in the text. Just to give one example: Supplementary Figure 1 h-k shown conditions of deleting sigma factors and the effect on NT presence. For each condition, proper quantification would show the mean number of nanotubes (say per 1000 cells), together with a standard deviation both calculated from multiple independent experiments, with the n number then used to also calculate a p-value for whether there is any effect relative to the control condition. Similar analysis can and should be done for almost every other statement made in the text. All values should be stated as a mean, with an error and an n-number and statements of comparison should have a p-value.

Another related concern is that the authors detect NTs as vary rare structures (just one NT for around 5000 cells) where they have previously been reported as common. There are two aspects to the problem - 1st could this be an issue with the way cells were cultured? Did the authors attempt to optimize growth conditions while checking for the presence of these structures etc? 2nd It raises concerns about how representative the images are. Presumably imaging large numbers of NTs to be representative by EM would therefore be very challenging? Do the authors have any

comment? The authors even state that some features of the Nile Red staining might be due to "non optimal conditions", but they do not elaborate on what these might be or why those conditions were not optimized.

The final major issue is I find it strange that so many nanotubes are found connecting two cells – here and in the literature. If they have no function and are driven purely by biophysical forces, these should be exceedingly rare coincidences. Could the authors comment?

Reviewer #4 (Remarks to the Author):

The study of Pospisil et al. studied bacterial nanotube formation concluding that they are formed through cell death. The formation and role of nanotube has been one of the exciting and controversial field in microbiology, and the authors approach in examining the state of the cells whether they are dead or alive are very important but that has been overlooked. The current manuscript show that cell death can lead to MV formation that seems to be universal among bacteria. One of the main concern is that , it is not clear whether the nanotubes that the authors observe are the same ones that have been reported by other groups, and the conclusions were not satisfactory justified. In the reviewer's opinion, they do not necessarily need to be the same and it may simply be that different routes for nanotube formation exist. The generic involvement of cell death in nanotube formation would add novel aspect on bacterial cell death on its own, if any biological function of these tubes can be addressed that is still missing. Also there are some concerns to be addressed and below are more specific comments.

1. The title and throughout the manuscript the authors challenge previous studies but the culture method that the authors used is unique, adding pressure to the cells, and relevant controls are missing to compare with previous studies. To confirm if the authors are indeed observing the same type of tube, they should at least include relevant controls such as the CORE mutant that was shown to reduce nanotube formation in earlier work (Bhattacharya et al, Cell Rep. 2019), and see if they produce less nanotube under their conditions. As mentioned above, it may simply be that several independent pathways exist for nanotube formation being cell death as one of them.
2. Line 86. The authors conclude that the cells produce low frequency of nanotubes compared to what have been reported. However, the previous reports use cells grown on a surface while the authors use liquid culture in their experiment. I think this makes a huge difference.
3. The reason why the authors applied pressure to the coverside is not explained and the biological significance of this method is unclear
4. The authors claim that the cell forming tubes are or become ghost cells in phase contrast image. While nanotube extending from ghost cells can be observed, it seems nanotube are also extending from intact cells, such as shown in Fig. 1C.
5. Fig.2. Looking at the SEM image in Fig. 2a the cells under pressure looks severely damaged showing sacs of peptidoglycan and some of cells with disrupted cell pole. These cells are not observed in previous studies and again it may be the consequence of the authors method that is producing these types of nanotubes.
6. Fig. 3. The decrease in SYTOX green signal intensity is defined as DNA release. Do the authors observe extracellular DNA?
7. Related to the experiment in Fig.4, in previous studies exchange of protein and mRNA by nanotube was examined but not in this study. Instead, they examined non-conjugative plasmid transfer that is another story, and cannot be compared to protein transfer or mRNA transfer. In line 190, the authors claim the sensitivity of their approach but I consider that plasmids are more "bulky" and would not go through the nanotube. The DnaseI treatment experiment support this by suggesting that the plasmids are not inside the nanotube but rather naked. Furthermore, and critically, the authors did not apply any pressure to the cells in these experiments, which in the beginning of the study was shown to form very few nanotubes under their condition used.

8. The authors detect cytoplasmic ZsGreen in their nanotubes and may use immunogold staining to examine their transfer among cells.
9. Is there any explanation why sigD deletion in the recipient influence the transformation efficiency in Fig. 4? Also the competence seems high in LB. To my knowledge, *B. subtilis* 168 do not show much competence under LB medium.
10. Fig. 5. The phase contrast image is not convincing enough that the cells are ghost cells. While SYTOX green and PI are generally used to indicate dead cells it is currently unclear whether nanotube formation alter the permeability of the cells. In addition, the influence of pressure is not examined here properly since the control without pressure is missing. To my experience, "burning" the cells with laser also results in nanotube formation. In addition, *E. coli* looks like they are producing vesicles rather than nanotubes.
11. Fig. 6. The role of pressure and autolysins are not discussed in this model, which I consider is the novel part of this study.

Dear Editor and Reviewers,

We thank the Reviewers for their stimulating comments!

A Summary of the changes and point-by-point responses follow. In the point-by-point responses we numbered the questions/comments to simplify orientation in the file.

The revised manuscript contains also all changes visible in the Track changes option. All references (page x, line y) in the following text to changes in the manuscript are done according to the final PDF file.

Summary of major changes:

- 1) We added a quantification of the phase contrast signal of dying cells into **Fig. 3c**.
- 2) We modified **Fig. 4c**. Now, Fig. 4c shows the correlation between transformation efficiency and *comK* expression.
- 3) We performed new microscopic experiments and also re-analyzed all our data. A statistician joined our team (MM, now one of the authors of the manuscript) and performed statistical analyses on the data. As a result, we not only provide the relevant statistical information in the manuscript (*e. g.* p-values) but we also show in detail all the statistical analyses in **Supplemental file 2**.

- 4) A new experiment with the LytEF double knock-out strain was performed (**Fig. S6d-e**).
- 5) We added **Supplemental movie 2** and **Fig. S5** showing events during the dying process.
- 6) We moderated the statements about cell death and we now use mostly “dying” instead of “dead” throughout the manuscript.
- 7) We created detailed cartoons of sample preparation methods to make the information more reader-friendly (**Fig. S16**).

Reviewer #1 (Remarks to the Author):

In their manuscript, Pospíšil and co-workers analyse the formation of bacterial structures termed "nanotubes". These poorly understood and controversial cell envelope-associated structures forming cell-to-cell contacts are implicated in inter- and cross species nutrient and DNA exchange. While these structures have been observed by few groups independently, there have also been substantial difficulties in reproducing the observations by many other groups. In this paper, Pospíšil and co-workers provide strong evidence that the formation of the nanotube-structures are highly dependent on the experimental conditions used, thus providing the first coherent explanation why the reproduction of the nanotube structures has been challenging. Furthermore, the authors provide compelling evidence that the formation of nanotubes is associated with the cell lysis process, and that the observed exchange of DNA is in fact facilitated by the *B. subtilis* competence machinery, rather than via nanotubes. Both these findings severely question the postulated models linking the observed nanotube-structures to specific biological functions. The study is highly timely and will be of considerable interest for those working on bacterial cell envelopes. While this manuscript will certainly not remain the "final word", it is a highly important contribution to a controversial topic. I therefore strongly endorse the publication of this manuscript in Nature Communications, subject to minor corrections listed below.

Minor concerns and corrections:

1) Lines 116-117:

Since the observation that cells producing nanotubes are losing phase contrast is quite central for this manuscript, a more thorough explanation of what "phase contrast" means is needed. I don't know what the authors mean with "loss of granularity". Please include an appropriate explanation how phase contrast relies on refractive index differences between cell interior and extracellular medium, loss of which indicating substantial leakage of cellular content.

RESPONSE

Live cells, full of cytoplasmic material can be distinguished from their environment by phase contrast imaging. This is based on differences between refractive indexes of live cells and extracellular space. Live cells have high values of refractive index and thus are visible as dark bodies. Ghost cells have lost their cellular content and this dramatically changes their refractive index. Consequently, ghost cells appear as greyish bodies that are even in some cases almost undetectable by phase contrast imaging. The same phenomenon (changes in phase contrast refractive index) was observed in both eukaryotic and prokaryotic cells in previous papers^{1,2} where this change was associated with cell dying.

ACTION TAKEN

We quantitated the changes in refractive index in phase images and correlated with the SYTOX Green signal. This correlation is now part of **Fig. 3c**. The phase image signal (black was defined as the highest signal, shades of grey as losing the signal) within the first ca 1 min (for wt) is inversely correlated with the SYTOX Green signal (an example of primary data is in **Fig. 3a**). The presence of the SYTOX signal in the cells indicates that they are dying, and the changes in their phase image refractive indexes indicate that they are losing their cytoplasmic contents. Such cells are on their way to become ghost cells – empty shells.

We describe the changes in phase contrast on **page 3, line 123**, and define ghost cells. Also, we added the above mentioned references.

"Loss of granularity" is defined as "loss of large proteins", indicative of leakage of the cell contents, incompatible with life³. Nevertheless, this is a fine point and we deleted "loss of granularity" from the manuscript.

2) Line 118 and throughout the manuscript:

While the authors rightly link the loss of phase contrast to cell death, the use of wording with respect to death and lysis is somewhat confusing throughout the manuscript. Both loss of phase contrast and cytoplasmic Sytox Green/Propidium iodide signals are perfectly valid indicators for cell lysis and, consequently, likely predictors for cell death. However, at which stage of the cell lysis process the cell in fact loose viability is unknown. In contrast, cells can very well loose viability without losing membrane integrity or phase contrast. Therefore, authors should appropriately explain the correlation with these two processes and, throughout the manuscript, refer to the cells loosing phase contrast/gaining Sytox Green signal as cell undergoing lysis, rather than over simplistically as "dead cells".

RESPONSE

We agree.

We realize that it is challenging to pinpoint the exact moment of cell death.

ACTION TAKEN

We substituted "dead" with "dying".

Moreover, to better describe the order of events, we added time-lapse microscopy movie of wt *B. subtilis* showing SYTOX green penetration, formation of NTs, DNA release, and ghost cell formation (**Supplemental Movie 2**). The movie shows originally darks cells in phase contrast. Then SYTOX green penetrates into these cells (e. i. the cell membrane must be impaired) and the appearance of the signal indicates binding of the dye to DNA. Subsequently, the SYTOX Green signal leaks out of the cell as DNA is released. This whole process is associated with a decrease in phase contrast of the cell and formation of ghost cells and nanotubes. Nanotubes always appear subsequent to SYTOX Green penetration. We refer to the movies in the main text on **page 4, line 169**. Finally, a description and scheme of **Supplemental Movie 2** were added into Supplementary data (**Supplementary Data Fig. 5**).

3) Line 139:

Sytox Green staining is not an indicator for impaired membrane potential. Rather, it is unable to diffuse across intact membranes and only stains cells with cells with large pores formed by antibiotic or proteins, or when the cell undergo lysis. Cells with low membrane potential do not, per se, become permeable for Sytox Green. Please correct appropriately.

RESPONSE

We agree.

Sytox Green staining is not an indicator for impaired membrane potential. SYTOX green indicates compromised/permeabilized membranes that allow its crossing into the cell and binding to DNA⁴.

ACTION TAKEN

We modified the text accordingly, **page 4, line 149**.

4) Figure 1 (and throughout the manuscript):

It is not clear what the authors mean with T=0 min. Assembling the slide, finding the cells under the microscope and imaging always take "some time" making 0 min certainly an incorrect value. Please provide a realistic estimate in the respective legends what the T=0 min represents in real terms (time from coverslip addition to image capture).

RESPONSE

t=0 min is the start of observation under the microscope. To prepare the sample and start the microscopy takes ca 30 seconds.

ACTION TAKEN

Definition of t=0 is now included in Materials and Methods section (**page 9, line 379**). A reference to this Mat & Met section (with respect to t=0) is now made in **Fig. 1** legend (**page 17, line 723**).

5) Figure 2A:

Please provide and overlay between the Nile Red and Sytox Green images. Replace life/dead cells with intact/lysed cells.

RESPONSE

a/ We created the requested **Fig. 2a** with the overlay (see **Reviewer's R-Fig. 1** at the end of this file). However, when printed, the panels are too small to see sufficient details.

b/ As already mentioned above, the life/death boundary is difficult to define. A correct way to group the cells could be SYTOX-positive/-negative but this would make the Figure less intuitive.

ACTION TAKEN

a/ For reasons discussed above, we prefer to keep **Fig. 2a** as it is. If the reviewer feels strongly about the overlay, we could add it to Supplementary data.

b/ We changed "dead" to "dying" and we kept "live" (**Fig. 2b, c**). However, we realize that "live" may also include dead AND intact cells (impenetrable to SYTOX). Therefore, we added this as a caveat to **Fig. 2** legend (**page 19, line 747**).

6) Line 144-145:

Poly-lysine penetrates bacterial membranes resulting in growth inhibition and loss of membrane potential (Katsu et al, Biochem Biophys Res Commun, 1984) making it perhaps not so surprising that poly-P aggravates the cell lysis process. Please mention the antibacterial, membrane-acting properties of poly-lysine in an appropriate place of the manuscript.

RESPONSE

We agree.

Nevertheless, poly-L-lysine is likely not a major causative agent of bacterial lysis in our samples. This is apparent in **Fig. S4b** where bacteria were spotted on poly-L-lysine coated glass slide and only partially covered with a poly-L-lysine coated coverslip. Only cells under the coverslip started massively dying but not the cells in their near vicinity. So, unless the presence of poly-L-lysine is required from both sides to induce cell dying, it seems that poly-L-lysine *per se* is not the main cause of cell death in our experimental setup.

ACTION TAKEN

We modified the text, added the reference, and the following sentence to **page 4, line 159**:

"This also shows that poly-L-lysine, which was previously reported to negatively impact bacterial growth⁵, is not a major factor causing SYTOX permeability."

7) Line 157:

It is not clear here if the authors mean nanotubes emerging within seconds of coverslip/pressure application, within second of Sytox Green entry, or something else. Please rephrase for clarity.

RESPONSE

We mean nanotubes emerging within seconds after SYTOX green entry.

ACTION TAKEN

We modified the text accordingly (**page 4, line 168**).

8) Figure 3A:

Please provide quantification of the cell Sytox Green signals, and the cell phase contrast signals (for instance as the delta of the average pixel inside vs outside of the cell) to accompany the individual microscopy images shown in panel A and B. This is to show if there really is a sudden increase in Sytox Green signal and phase contrast associated with the emergence of the nanotubes. This is not really clear from the images themselves.

RESPONSE

This issue is already addressed in our response to the first comment of Reviewer #1.

ACTION TAKEN

We now provide phase contrast quantifications of wt and $\Delta sigD$ images. The results are showed in **Fig. 3C** (open circles).

9) Lines 179-180 and Figure S7:

-Please include the phase contrast images for the cells shown in panels S7A-D.

RESPONSE

These images were taken with a CellR microscope. Our CellR is equipped with a cultivation chamber, which is most suitable for time lapse experiments of live cells on agar at optimal growth temperature. However, our CellR is not equipped with a phase contrast objective, only with a differential interference contrast objective (DIC). DIC allows to distinguish between dying and intact/live cells but less clearly than phase images.

ACTION TAKEN

DIC images were added next to respective fluorescence images. Also, as requested by you and Reviewer#3, we also changed the presentation of the quantitation graphs. Both these actions increased the size of the Figure - formerly Fig. S7. Now, it is in **Figs. S8, S9, S11 and S12**.

10) The membrane batches shown to emerge in Chloramphenicol and Rifampicin-treated cells (highlighted with arrows) look very much distinct from those observed with Ampicillin or Poly-P. I am not convinced such intracellular or membrane-associated patches can by any means be interpreted as nanotubes. Please re-analyse and strictly limit the quantification to structures extending away from the cell surface only. This will likely also need some toning down the statement in lines 179-182.

RESPONSE

This is a good point!

ACTION TAKEN

We re-examined and re-analyzed NTs on agar dishes (+/- antibiotic) as requested. The strict re-analysis revealed that in the case of "Cm" and "Rif", the number of NTs decreased while in the cases of "Amp" and "No antibiotic" the results remained unchanged. We altered also the way the data are presented to make the **Figs. S8, S9, S11 and S12** more reader-friendly. Finally, we modified the main text accordingly (**page 5, line 191**).

11) Lines 244-248:

-As in many other place in the manuscript, please do not refer to these cells as "dead". While Sytox Green is commercially sold as part of "life-dead staining", this more a sales pitch than a valid scientific statement. It is strictly a reporter for membrane integrity, not cell viability.

RESPONSE

We agree.

SYTOX green actually underestimates the number of dead cells⁴. Nevertheless, cells with compromised membranes are in severe difficulties, likely dying.

ACTION TAKEN

We altered the wording and refer to these cells as 'dying'. Please, see also our response to your comment about **Fig. 2a**.

12) I am not convinced by the author's interpretation that increased Nile Red staining indicates an ability to cross the membrane and stain the "inner leaflet". Nile Red is commonly used to stain intracellular fatty acid droplets in eukaryotes, and is perfectly able to stain Bacillus endospore membranes while still encased by the mother cell membranes. Both are strong indications that Nile Red can diffuse across intact membranes. I would omit this likely wrong und unnecessary speculation altogether.

RESPONSE

We agree with the reviewer about Nile Red. Nevertheless, we observed a similar effect also with FM4-64 dye, which intercalates only into the outer surface of biological membranes but is unable to cross the lipid bilayer, therefore only membrane surfaces directly exposed to FM4-64 will stain. During sporulation, both the mother cell and forespore membranes are accessible to FM4-64 until the engulfment of the forespore is completed. Once engulfment and membrane fusion are complete, the two membranes surrounding the forespore are isolated from the external environment. Thus, at the completion of engulfment, only the mother cell cytoplasmic membrane stains with FM4-64, while the forespore membranes remain unstained⁶. Taken together, probably our statement is not wrong but we agree with the reviewer that the statement is unnecessary speculation.

ACTION TAKEN

We removed this speculation.

13) Lines 267-268:

-The curvature dependency of Cardiolipin described in reference 39 (Beltran-Heredia et al.) refers to membrane curvatures order of a magnitude (and more) stronger than those observed at bacterial cell poles. While such a mechanism could well be involved in the extrusion of the thin, tubular nanotube-structures, I cannot follow the logic how this could explain the preferential emergence from cell poles (especially given the localization of Cardiolipin at B. subtilis cell poles has recently been questioned (Pogmore et al, 2018). Please rephrase.

RESPONSE

We agree.

ACTION TAKEN

We rephrased the text, see **page 6, line 277**.

14) Also, lipids don't "bind" curved membranes but localize to/enrich at curved membrane areas.

RESPONSE

We agree.

ACTION TAKEN

We deleted this part of the sentence.

15) Lines 292-302:

Given the very different composition, architecture and the cellular context, it is exceedingly likely that the mechanisms governing the formation of outer membrane tubes and inner membrane "nanotubes" are distinct. Therefore, discussing the observations of this paper in the context of OMTs seems a bit speculative. This is especially given it is unlikely that an outer membrane is affected from cell wall-linked lysis the same way as the inner membrane is. Please provide a more nuanced discussion here.

RESPONSE

We agree.

ACTION TAKEN

We expanded this part of Discussion, pointing out that OMTs are likely formed differently from *B. subtilis* NTs (**page 7, line 321**).

Reviewer #2 (Remarks to the Author):

The manuscript by Prospisil et al critically examines a striking and enigmatic phenomenon whereby bacteria, most notably *Bacillus subtilis*, form long narrow membrane filaments by an unknown mechanism. These structures, called nanotubes (NTs), received considerable recognition since a landmark publication almost a decade ago (ref 1). That publication raised skepticism in the field about claims that NTs indiscriminately form cell-cell conduits between cells of the same species and distantly related species wherein macromolecular content in the cytoplasm is exchanged. Those and other concerns were partly addressed in subsequent publications (e.g. refs 9, 11 and 15). However, a lingering concern was that a single lab did all of the work and that an independent lab had not corroborated the rather revolutionary claims (refs 1, 9, 15, 16). Therefore, this work by Prospisil et al, is important because they are the first outside group to examine *Bs* NTs. These authors convincingly show that *Bs* NT production is caused by cell stress/death leading to punctures in the cell wall and the cytoplasmic membrane being extruded as NTs. These findings were shown by a variety of fluorescent and EM techniques using different reporters and mutant strains. They also demonstrate that cell-cell transfer of cytoplasmic plasmids are not mediated by NTs, as previously claimed, but instead occurs by donor cell lysis and DNA uptake by natural competence in the recipient cell. This work is carefully done, well written and the findings will have a broad impact to the microbiology field and will stimulate future research.

Specific comments:

1) Line 35: The first paragraph is a bit murky as different membrane tube structures are discussed as if they are the same, which they are not. Possible solution, change "These structures may serve different functions in these different organisms" to "These structures are composed of cytoplasmic or outer membranes, depending on the species of origin, and may serve different functions in these organisms."

RESPONSE

We agree.

ACTION TAKEN

We have made the requested change (**page 1, line 39**).

2) Line 44: For clarity, please change “types” to “configurations”

RESPONSE

We agree. However, “configuration” does not sound as the right word.

ACTION TAKEN

We have substituted “types” with “classes” (**page 2, line 48**).

3) How do the authors know the narrow (red arrow) filaments in Sup. Fig. 1 SEMs (and elsewhere) are not exopolysaccharide (EPS) bundles or filaments.

RESPONSE

These filamentous structures are present in samples from cells that form flagella. Samples from *e. g. Δhag* flagella-less cells completely lack these structures as seen in **Fig. S1b, c**. Therefore, it is highly likely that these structures are flagella.

4) Line 88-9: Please discuss why the frequency of NTs is dramatically lower (1/5,000 cells) compared to prior reports. Prior studies found that NTs were primarily made by cells in biofilms. Here the investigators used planktonic cells. Are there more NTs when cells are grown as biofilms?

RESPONSE

The numbers of NTs identified in previous studies are speculative. Most of the quantification experiments in these studies were done by SEM, which cannot, on its own, discriminate between nanotubes and exopolysaccharide bundles. Therefore, these studies grossly overestimate the number of NTs. This is the case when the cells are grown on solid media; when the cells are grown in liquid media, the number of cell-to-cell “connections” is much lower due to the absence of exopolysaccharide bundles.

The correct way to visualize and count NTs is by fluorescence microscopy that allows to detect tubular membranous structures.

ACTION TAKEN

We re-analyzed our data and stringently counted NTs in our SEM and SIM experiments. Also, to the resulting data sets, we applied statistical analyses as requested by Reviewer #3. Most importantly, these data are both for planktonic cells and cells grown in biofilms.

We added **Fig. S16** that together with **Fig. S17** (formerly Fig. S12) reveals the difference between samples grown in liquid and on solid media. Cartoons of sample preparation techniques are shown in **Fig. S16a, e**. **Fig. S16b-d** show no tubular structures both in SIM and SEM images of planktonic cells (*Δhag* cell were used for the SEM image to remove flagella). Next, **Fig. S16f-h** shows the same type of images for solid media-grown cells. No NT were detected in **Fig. S16f** while numerous tubular structures (likely exopolysaccharide bundles) were identified in **Fig. S16g** (green arrows). These results strongly suggest that the previously observed tubular structures were most likely not membranous NTs. A paragraph commenting on this issue was added to Discussion (**page 6, line 281**).

5) The authors should incorporate a very recent publication into their discussion (e.g. line 129): Baidya, Rosenshine & Ben-Yehuda. Donor-delivered cell wall hydrolases facilitate nanotube penetration into recipient bacteria. 2020, Nat Comm.

RESPONSE

We agree.

ACTION TAKEN

This publication is now mentioned in a sentence in Discussion and cited on **page 6, line 256**.

6) Line 140: SYTOX is stated twice in one sentence.

RESPONSE

The second SYTOX (in parenthesis) is an abbreviation for "SYTOX green stain" so that we would not have to use this long descriptive name in subsequent text.

7) The experiments in Figure 4 nicely and critically show that cell-cell transfer occurs via naked DNA and natural competence; not mediated by NTs. However, the transfer assay was done in liquid. In prior studies, e.g. ref 1, transfer was done when cell were spotted on agar (biofilm-like). What happens if the experiments are instead done on agar? This is relevant because DNA transfer mechanisms could be fundamentally different between liquid and solid media.

RESPONSE

We agree. In Dubey et al.⁷ the cells were co-incubated for 4 h on solid media without any antibiotics and then replica-plated on agar dishes supplemented (or not) with appropriate antibiotics. However, in subsequent studies^{8,9} performed by the same group, a liquid media approach was used. By this approach they claim to see plasmid transfer with the same qualitative results. We use the same liquid media approach and, additionally, provide also quantitative evaluations of the experiments that unambiguously show that plasmid transfer is strictly dependent on competence of the cells. Moreover, **Figs. S16, S17** reveal a lack of NTs formed by unstressed cells grown on agarose pads, strongly arguing against the possibility of plasmid transfer by these structures on solid media.

8) Fig. 4B legend needs clarification in context of "appropriate antibiotics." It seems in every case at least two antibiotics were added and perhaps in some cases three?

RESPONSE

Yes.

ACTION TAKEN

We added a more thorough description of the antibiotic combinations into the **Fig. 4b** legend (**page 23, line 793**).

9) The legend to Supp Fig. 5a is not clear. The bar graph seems to show percentage of cells with NTs, but, for example, what is the difference between "Live cells" and "Live cells NTs?" Also, in panel B, why are the cells bright green (SYTOX positive) at t = 0?

RESPONSE

We agree.

ACTION TAKEN

We completely changed the graphs in **Fig. S6a-b** and the legends were changed accordingly. Moreover, charts for the Δ ytEF strain were added (**Fig. S6d-e**).

10) Supp Fig. 7/legend. No “white arrows” are shown; only yellow arrows. Please correct. In the bar graphs what does “Percentage” represent, e.g. in “Live cells NT”? It sounds like percentage of cells with NTs that have NTs.

RESPONSE

Actually, there was a white arrow in **Fig. S7Da** at 90 min. However, the arrow was small and easy to overlook.

ACTION TAKEN

The original Fig. S7 exists no more. Now, the information is divided between **Figs. S8, S9, S11** and **S12**. The new Figures contain the original information from the original Fig. S7 plus DIC fields of respective images and quantitative analysis presented by box plots.

11) Supp Fig. S9. Add white arrow(s) to panel C.

ACTION TAKEN

A white arrow was added. The original Fig. S9 is now **Fig. S13**.

12) Line 297-99: These sentences are not accurate. Change “...phase-contrast images showing that it is predominantly ‘ghost’ cells that form OMTs” to “phase-contrast images suggest ‘ghost’ cells could contribute to the formation of OMTs.”

RESPONSE

We agree. Nevertheless, we decided to change the wording (slightly different from the suggested one) as ghost cells likely do not contribute to OMT formation. Rather, ghost cells might be a consequence of OMT formation because OMT formation possibly leads to release of the cytoplasmic contents and this, ultimately, results in a ghost cell.

ACTION TAKEN

We replaced the sentence with:

“...phase-contrast images suggesting that it is predominantly ‘ghost’ cells that are associated with OMTs.” (**page 7, line 316**).

13) Also change “The authors did not discuss this phenomenon but those cells were most likely dead.” to “Although these authors discussed OMT formation was caused by cell stress they did not emphasize that some cells might be dead.”

RESPONSE

We agree.

ACTION TAKEN

We replaced the original sentence with the following sentence:

“Although these authors discussed that OMT formation was caused by cell stress they did not explore the possibility that some cells were dying.” (**page 7, line 317**).

14) Figure 6 nicely summarizes the findings in the paper.

RESPONSE

Thank you!

Reviewer #3 (Remarks to the Author):

Here, Pospíšil et al investigate membranous nanotubes previously observed connecting bacterial cells. While previous work has hypothesised functional roles for these structures, Pospíšil et al conclude that these are manifestations of cell death, formed from purely biophysical forces and are of no functional significance. If that conclusion is correct, then the result is indeed interesting and I would recommend acceptance in Nature Communications. That said, there is a major concern with the validity of the results because no statistical analysis is ever presented. If the statements made were backed up by properly calculated stats and p-values I would recommend publication. See below for details and other major concerns. Until those are in place, it's basically impossible to adequately review the manuscript.

1) The major issue is that there are no statistics. No p-values are ever shown for any of the statements made in the text. Just to give one example: Supplementary Figure 1 h-k shown conditions of deleting sigma factors and the effect on NT presence. For each condition, proper quantification would show the mean number of nanotubes (say per 1000 cells), together with a standard deviation both calculated from multiple independent experiments, with the n number then used to also calculate a p-value for whether there is any effect relative to the control condition. Similar analysis can and should be done for almost every other statement made in the text. All values should be stated as a mean, with an error and an n-number and statements of comparison should have a p-value.

RESPONSE

We agree. Although we believe that even without additional statistics the study demonstrates the point, providing more detailed statistical analysis can only help.

ACTION TAKEN

We included a statistician into our team (Dr. M. Modrák) who is now also a coauthor of the manuscript. Moreover, we conducted new microscopy experiments, re-analyzed all images and stringently counted nanotubes/tubular structures and performed statistical analyses on the data. Now, both in the text and in Figure legends, p-values are provided wherever applicable. We further included a paragraph into the Mat & Met section, describing the statistical analysis. Finally, we provide an additional supplementary file (**Supplemental file 2**), describing in detail all the statistical analyses performed in this study.

2) Another related concern is that the authors detect NTs as very rare structures (just one NT for around 5000 cells) where they have previously been reported as common. There are two aspects to the problem - 1st could this be an issue with the way cells were cultured? Did the authors attempt to optimize growth conditions while checking for the presence of these structures etc.? 2nd It raises concerns about how representative the images are. Presumably imaging large numbers of NTs to be representative by EM would therefore be very challenging? Do the authors have any comment? The authors even state that some features of the Nile Red staining might be due to "non optimal conditions", but they do not elaborate on what these might be or why those conditions were not optimized.

RESPONSES

1/ The cells were cultured in 10 mL of LB in 100 mL Erlenmeyer flasks. The temperature was 37 °C. **Fig. S3b** shows the growth curve, the doubling time was typically around 20 min - nice, vigorous exponential growth. The cells were typically withdrawn for further analysis in midlog phase ($OD_{600} \sim 0.6$). When the cells were viewed by fluorescence microscopy, the number of NTs was relatively low. These NTs originated most likely from dying cells.

We also sought conditions when NTs would be more numerous. We found such conditions - when pressure was applied and when stressors, such as ampicillin, were added.

We also compared liquid and solid media (see next point) and, in fact, more NTs can be detected from cells grown in liquid media, likely due to all the manipulations with the sample (centrifuging,

staining). When the cells are grown on solid media, undisturbed (**Fig. S16f and h**), no NTs are detected by membrane staining.

2/ How representative are the images shown in our Figures? As finding NTs in unstressed cells by fluorescence microscopy is challenging, we show images where an NT is visible. The remaining parts of the images, therefore, contained (on average) fewer NT than the selected part of the image! Typical images of full fields (SIM and SEM) are shown in **Figs. S1 and S16**.

Finally, we strongly believe that the previously reported high counts of NTs might stem from e.g. mistaking exopolysaccharide bundles for NTs (as observed by SEM). Therefore, the most convincing method to detect NTs is fluorescence microscopy.

ACTION TAKEN

As described above, we re-analyzed all the images, both from electron and fluorescence microscopy. In SEM images, it is difficult to distinguish between NTs and other tubular structures. Combined with what we see in fluorescence microscopy (where membranous tubular structures can be unambiguously identified), we realized that, most likely, the thick, rope-like structures observed in SEM images are NTs. If we counted ALL the tubular structures, we would arrive at a significantly higher number. However, this number would not reflect the true counts of NTs.

Our stringent re-analysis, both by SEM and SIM, of images from LB-grown exponential cells yielded less than 1 % of cells with NTs (**Fig. S11**). Most importantly, **Fig. S16** shows comparisons of different methods of (i) culturing the cells (liquid or solid media), (ii) sample preparation, and (iii) observing the cells. **Figs. S16b-c** show no tubular structures both in SIM and SEM images of planktonic cells (Δhag cell were used for the SEM image to remove flagella). Next, **Fig. S16f-g** show the same type of images for solid media-grown cells: no NTs were detected in **Fig. S16f** while numerous tubular structures (likely exopolysaccharide bundles) were identified in **Fig. S16g**. The number of these structures approaches the reported numbers of "NTs" as is shown in **Fig. S16h**. These results strongly suggest that the previously observed tubular structures were most likely not membranous NTs. A paragraph commenting on this issue was added to Discussion (**page 6, line 281**).

3) The final major issue is I find it strange that so many nanotubes are found connecting two cells – here and in the literature. If they have no function and are driven purely by biophysical forces, these should be exceedingly rare coincidences. Could the authors comment?

RESPONSE

As described above, structures marked with green arrows in **Figs. S16 and S17** are not NTs. By membrane staining (SIM images – **Fig. S16f**, quantification – **Fig. S16h**), no cell-to-cell connecting NTs were detected.

Reviewer #4 (Remarks to the Author):

The study of Pospisil et al. studied bacterial nanotube formation concluding that they are formed through cell death. The formation and role of nanotube has been one of the exciting and controversial field in microbiology, and the authors approach in examining the state of the cells whether they are dead or alive are very important but that has been overlooked. The current manuscript show that cell death can lead to MV formation that seems to be universal among bacteria. One of the main concern is that, it is not clear whether the nanotubes that the authors observe are the same ones that have been reported by other groups, and the conclusions were not satisfactory justified. In the reviewer's opinion, they do not necessarily need to be the same and it may simply be that different routes for nanotube formation exist. The generic involvement of cell death in nanotube formation would add novel aspect on bacterial cell death on its own, if any biological function of these tubes can be addressed that is still missing. Also there are some concerns to be addressed and below are more specific comments.

1) The title and throughout the manuscript the authors challenge previous studies but the culture method that the authors used is unique, adding pressure to the cells, and relevant controls are missing to compare with previous studies. To confirm if the authors are indeed observing the same type of tube, they should at least include relevant controls such as the CORE mutant that was shown to reduce nanotube formation in earlier work (Bhattacharya et al, Cell Rep. 2019), and see if they produce less nanotube under their conditions. As mentioned above, it may simply be that several independent pathways exist for nanotube formation being cell death as one of them.

RESPONSE

Our data show that undisturbed, solid media-grown cells do not form NTs (**Fig. S17a-b**). Moreover, **Fig. S16** (and responses to Reviewer #3, second comment) dissect three variables of NT (or NT-like structures) formation and present a comprehensive view of this phenomenon.

Moreover, we identified the SigD regulon as required for the early NT formation as opposed to its deletion strain where NT formation is delayed (**Fig. 3** and **Fig. S6a-b**). This correlates with the CORE mutant (lacking genes). The loss of these genes leads to inactivation of SigD by preventing FlgM export and proteolysis¹⁰.

ACTION TAKEN

We added **Fig. S16**. We added the information about CORE, SigD, and FlgM into Discussion (**page 6, line 251**).

2) Line 86. The authors conclude that the cells produce low frequency of nanotubes compared to what have been reported. However, the previous reports use cells grown on a surface while the authors use liquid culture in their experiment. I think this makes a huge difference.

RESPONSE

Please, see our 2nd response to Reviewer #3.

3) The reason why the authors applied pressure to the coverside is not explained and the biological significance of this method is unclear

RESPONSE

The application of some pressure is part of the sample preparation process – this serves to create a monolayer of cells, which facilitates sample observation. That is actually how we discovered this phenomenon. As for biological explanation, we interpret the data so that NTs are formed by dying/dead cells and their biological role is unclear (if any).

ACTION TAKEN

We modified the text (**page 3, line 115**). We now provide a brief description of how the role of pressure was discovered.

4) The authors claim that the cell forming tubes are or become ghost cells in phase contrast image. While nanotube extending from ghost cells can be observed, it seems nanotube are also extending from intact cells, such as shown in Fig. 1C.

RESPONSE

We agree. Ghost cells are the final stage, the empty shells. NTs are the first outbursts of the cytoplasmic content (encased within the nanotube) from the cell. This is followed by more extensive damage to the cell envelope that allows chromosomal DNA to leak out. What remains is the ghost cell to which NT may be attached.

ACTION TAKEN

To illustrate the order of events we included **Supplemental Movie 2**. The movie is linked with **Fig. S5**, where description/interpretation of the events is provided.

5) Fig.2. Looking at the SEM image in Fig. 2a the cells under pressure looks severely damaged showing sacs of peptidoglycan and some of cells with disrupted cell pole. These cells are not observed in previous studies and again it may be the consequence of the authors method that is producing these types of nanotubes.

RESPONSE

You are right. **Fig. 2a** shows what type of damage is caused by the "pressurized" type of sample preparation for fluorescence microscopy.

ACTION TAKEN

We added a sentence explaining this issue to **page 4, line 151**:

"The SEM images illustrate the severe damage caused to cells by the P-GLG experimental setup."

6) Fig. 3. The decrease in SYTOX green signal intensity is defined as DNA release. Do the authors observe extracellular DNA?

RESPONSE

Yes, we do. In movies we observe DNA release visible as efflux of the DNA-bound SYTOX green stain.

ACTION TAKEN

We added a supplemental movie illustrating this point (**Supplemental Movie 2** and **Fig. S5**; see also our response to the second comment of Reviewer #1).

7) Related to the experiment in Fig.4, in previous studies exchange of protein and mRNA by nanotube was examined but not in this study. Instead, they examined non-conjugative plasmid transfer that is another story, and cannot be compared to protein transfer or mRNA transfer. In line 190, the authors claim the sensitivity of their approach but I consider that plasmids are more "bulky" and would not go through the nanotube. The DNase I treatment experiment support this by suggesting that the plasmids are not inside the nanotube but rather naked. Furthermore, and critically, the authors did not apply any pressure to the cells in these experiments, which in the beginning of the study was shown to form very few nanotubes under their condition used.

RESPONSE

Yes, plasmids are bulkier than proteins. However, they offer superior detection sensitivity. Nevertheless, the plasmid that we used in our experiments, pCPP31-Y1 (5.6 kb)¹¹, is even smaller than pHB201 (6.6 kb), which had been used in previous reports⁷.

We also attempted (quite extensively) to detect protein transfer but without success. Please, see also our response to your next comment.

Finally, there was no reason to use pressure (in other words, kill the cells) in our plasmid transfer experiments. This would dramatically decrease the cell counts, and make the experiment difficult to interpret. We wanted to study plasmid transfer between live cells, in experimental settings similar/identical to those already published^{8,9}.

8) The authors detect cytoplasmic ZsGreen in their nanotubes and may use immunogold staining to examine their transfer among cells.

RESPONSE

Yes, it is possible. We tried to detect transfer of the ZsGreen protein between cells by FACS and GFP by fluorescence microscope. However, despite our best efforts we were not able to see any transfer (data not shown). Immunogold, from our point of view, could be a tricky approach with a potential for artefacts.

9) Is there any explanation why sigD deletion in the recipient influence the transformation efficiency in Fig. 4?

RESPONSE

The *hag* gene, which is strictly SigD-dependent, is known to affect competence. Deletion of the *hag* gene has a negative effect on competence¹², consistent with our result. Moreover, we show in **Fig. 4c** that also the level of the *comK* mRNA is decreased in the SigD knock-out strain, likely further contributing to the decreased competence of the strain.

ACTION TAKEN

We added the mentioned reference to the text and included a brief explanatory sentence (**page 5, line 227**).

We created a new figure that shows the correlation between *comK* mRNA levels and transformation efficiencies. This Figure is now **Fig. 4c**. The original Fig. 4c was removed as redundant.

Also the competence seems high in LB. To my knowledge, *B. subtilis* 168 do not show much competence under LB medium.

RESPONSE

We agree. *B. subtilis* cells are less competent in LB than under low nutrition conditions that massively induce this physiological state. Nevertheless, *B. subtilis* grown in LB can be transformed both with plasmid and chromosomal DNA¹³ and the transformation efficiency (in LB) increases when *comK* is overexpressed, the same trend as we observe.

ACTION TAKEN

We added the mentioned reference to the text and included a brief explanatory sentence (**page 5, line 229**).

10) Fig. 5. The phase contrast image is not convincing enough that the cells are ghost cells. While SYTOX green and PI are generally used to indicate dead cells it is currently unclear whether nanotube formation alter the permeability of the cells. In addition, the influence of pressure is not examined here properly since the control without pressure is missing. To my experience, “burning” the cells with laser also results in nanotube formation. In addition, *E. coli* looks like they are producing vesicles rather than nanotubes.

RESPONSE

These experiments (**Fig. 5**) are an extension of the *B. subtilis* experiments, using only the conditions that we found to induce massive NT formation. Although we do not have the “no pressure” control, we show images at t=0 where no NTs/blebs were detected and the cells were still SYTOX negative.

We do not claim that the cells at the later time points were ghost cells but they were SYTOX positive and likely dying. Nevertheless, changes in refractive index in *B. megaterium* and *E. coli* were visible. Finally, we state in the manuscript: “*E. coli* reacted mostly by forming blebs, or vesicles, and occasionally also NTs.” (**page 6, line 241**), which is in line with the Reviewer’s observation.

11) Fig. 6. The role of pressure and autolysins are not discussed in this model, which I consider is the novel part of this study.

RESPONSE

We agree.

ACTION TAKEN

We altered **Fig. 6**, and we mention now the involvement of autolysins in NT formation on **page 6, line 252**.

R-Fig. 1. This Figure is almost identical with Fig. 2a in the manuscript. In addition, it contains the Merge column. For more information, see Response to Comment 5/ of Reviewer #1.

References

- 1 Lee, S. *et al.* Dynamic analysis of pathogen-infected host cells using quantitative phase microscopy. *J Biomed Opt* **16**, 036004, doi:10.1117/1.3548882 (2011).
- 2 Mohamed, Y. F. & Valvano, M. A. A Burkholderia cenocepacia MurJ (MviN) homolog is essential for cell wall peptidoglycan synthesis and bacterial viability. *Glycobiology* **24**, 564-576, doi:10.1093/glycob/cwu025 (2014).
- 3 Reuter, M. *et al.* Mechanosensitive channels and bacterial cell wall integrity: does life end with a bang or a whimper? *J R Soc Interface* **11**, 20130850, doi:10.1098/rsif.2013.0850 (2014).
- 4 Roth, B. L., Poot, M., Yue, S. T. & Millard, P. J. Bacterial viability and antibiotic susceptibility testing with SYTOX green nucleic acid stain. *Appl Environ Microbiol* **63**, 2421-2431 (1997).
- 5 Katsu, T., Tsuchiya, T. & Fujita, Y. Dissipation of membrane potential of Escherichia coli cells induced by macromolecular polylysine. *Biochem Biophys Res Commun* **122**, 401-406, doi:10.1016/0006-291x(84)90489-3 (1984).
- 6 Sharp, M. D. & Pogliano, K. An in vivo membrane fusion assay implicates SpoIIIE in the final stages of engulfment during Bacillus subtilis sporulation. *Proc Natl Acad Sci U S A* **96**, 14553-14558, doi:10.1073/pnas.96.25.14553 (1999).
- 7 Dubey, G. P. & Ben-Yehuda, S. Intercellular nanotubes mediate bacterial communication. *Cell* **144**, 590-600, doi:10.1016/j.cell.2011.01.015 (2011).

- 8 Baidya, A. K., Rosenshine, I. & Ben-Yehuda, S. Donor-delivered cell wall hydrolases facilitate nanotube penetration into recipient bacteria. *Nat Commun* **11**, 1938, doi:10.1038/s41467-020-15605-1 (2020).
- 9 Bhattacharya, S. *et al.* A Ubiquitous Platform for Bacterial Nanotube Biogenesis. *Cell Rep* **27**, 334-342 e310, doi:10.1016/j.celrep.2019.02.055 (2019).
- 10 Calvo, R. A. & Kearns, D. B. FlgM is secreted by the flagellar export apparatus in *Bacillus subtilis*. *J Bacteriol* **197**, 81-91, doi:10.1128/JB.02324-14 (2015).
- 11 Krasny, L., Vacik, T., Fucik, V. & Jonak, J. Cloning and characterization of the str operon and elongation factor Tu expression in *Bacillus stearothermophilus*. *J Bacteriol* **182**, 6114-6122, doi:10.1128/jb.182.21.6114-6122.2000 (2000).
- 12 Holscher, T. *et al.* Impaired competence in flagellar mutants of *Bacillus subtilis* is connected to the regulatory network governed by DegU. *Environ Microbiol Rep* **10**, 23-32, doi:10.1111/1758-2229.12601 (2018).
- 13 Rahmer, R., Morabbi Heravi, K. & Altenbuchner, J. Construction of a Super-Competent *Bacillus subtilis* 168 Using the P mtlA -comKS Inducible Cassette. *Front Microbiol* **6**, 1431, doi:10.3389/fmicb.2015.01431 (2015).

REVIEWERS' COMMENTS:

Reviewer #1 (Remarks to the Author):

The authors have done a very good job in addressing the various concerns I raised in the first round of reviews. The changes to the manuscript have made it significantly stronger and more quantitative, and overall a more polished manuscript. I am therefore happy to endorse the publication.

Reviewer #2 (Remarks to the Author):

I have reviewed the revised manuscript by Pospíšil et al on bacterial NTs. The authors have carefully and thoroughly addressed all concerns raised by prior reviews, including concerns raised about statistical analysis. I have no further concerns.

This manuscript critically examines a high profile and controversial topic in microbiology and provides strong evidence that NT production is the result of cell death in *Bacillus subtilis*. This work is of broad interest and importance and should be accepted for publication.

Reviewer #3 (Remarks to the Author):

The authors have performed a significant amount of extra work and in particular have addressed my concerns about the lack of statistics. I think the manuscript is now acceptable for publication.

Reviewer #4 (Remarks to the Author):

The authors did a great job in improving the manuscript. It is still unclear from the data whether the NTs the authors have examined are the same ones as observed in previous studies but I think the authors work would greatly contribute to advance the ongoing discussion in the field by giving different explanations of the NTs.

REVIEWERS' COMMENTS:

Reviewer #1 (Remarks to the Author):

The authors have done a very good job in addressing the various concerns I raised in the first round of reviews. The changes to the manuscript have made it significantly stronger and more quantitative, and overall a more polished manuscript. I am therefore happy to endorse the publication.

Thank you!

Reviewer #2 (Remarks to the Author):

I have reviewed the revised manuscript by Pospíšil et al on bacterial NTs. The authors have carefully and thoroughly addressed all concerns raised by prior reviews, including concerns raised about statistical analysis. I have no further concerns.

This manuscript critically examines a high profile and controversial topic in microbiology and provides strong evidence that NT production is the result of cell death in *Bacillus subtilis*. This work is of broad interest and importance and should be accepted for publication.

Thank you!

Reviewer #3 (Remarks to the Author):

The authors have performed a significant amount of extra work and in particular have addressed my concerns about the lack of statistics. I think the manuscript is now acceptable for publication.

Thank you!

Reviewer #4 (Remarks to the Author):

The authors did a great job in improving the manuscript. It is still unclear from the data whether the NTs the authors have examined are the same ones as observed in previous studies but I think the authors work would greatly contribute to advance the ongoing discussion in the field by giving different explanations of the NTs.

Thank you!